# Aligning Language Models with Demonstrated Feedback

**Omar Shaikh**[*]
Stanford University
oshaikh@stanford.edu

**Michelle S. Lam**[*]
Stanford University
mlam4@cs.stanford.edu

**Joey Hejna**[*]
Stanford University
jhejna@cs.stanford.edu

**Yijia Shao**
Stanford University

**Hyundong Cho**
USC

**Michael S. Bernstein**
Stanford University

**Diyi Yang**
Stanford University

## Abstract

Language models are aligned to emulate the collective voice of many, resulting in outputs that align with no one in particular. Steering LLMs away from generic output is possible through supervised finetuning or RLHF, but requires prohibitively large datasets for new ad-hoc tasks. We argue that it is instead possible to align an LLM to a specific setting by leveraging a very small number ($< 10$) of demonstrations as feedback. Our method, Demonstration ITerated Task Optimization (DITTO), directly aligns language model outputs to a user's demonstrated behaviors. Derived using ideas from online imitation learning, DITTO cheaply generates online comparison data by treating users' demonstrations as preferred over output from the LLM and its intermediate checkpoints. Concretely, DITTO operates by having an LLM generate examples that are presumed to be inferior to expert demonstrations. The method iteratively constructs pairwise preference relationships between these LLM-generated samples and expert demonstrations, potentially including comparisons between different training checkpoints. These constructed preference pairs are then used to train the model using a preference optimization algorithm (e.g. DPO). We evaluate DITTO's ability to learn fine-grained style and task alignment across domains such as news articles, emails, and blog posts. Additionally, we conduct a user study soliciting a range of demonstrations from participants ($N = 16$). Across our benchmarks and user study, we find that win-rates for DITTO outperform few-shot prompting, supervised fine-tuning, and other self-play methods by an avg. of 19% points. By using demonstrations as feedback directly, DITTO offers a novel method for effective customization of LLMs.[1]

## 1 Introduction

Large language models (LLMs) are trained for general-purpose use. In practice, however, they are often applied to very specific tasks for very specific users. Consider a task as simple as writing an email: our preferred email depends on personal writing style, the specific email task, or the target audience (a friend, stranger, etc.). As a result, there can be a mismatch between the universal style (Santurkar et al., 2023; Chakrabarty et al., 2023) trained into an LLM via instruction and preference tuning, and the specific style needed for applications. LLM outputs feel unopinionated and generic because of this mismatch.

While existing approaches such as supervised or preference finetuning are effective, they can require a large corpus of (un)acceptable behavior (on the order of $\approx 1K$ samples (Zhou et al., 2024; Ouyang et al., 2022)), which in turn requires unreasonably high effort from an individual. RLAIF methods like Constitutional AI (Bai et al., 2022) automate pairwise preference collection with an LLM, but align models to general principles that may not capture fine-grained preferences. Although prompting is data efficient, finding an effective prompt can be tedious—end-users often rely on brittle prompting heuristics (Zhou et al., 2022; Zamfirescu-Pereira et al., 2023). How might we efficiently communicate preferences and align a language model to a new individual or task?

---

[*]Equal Contribution
[1]Code: https://github.com/SALT-NLP/demonstrated-feedback

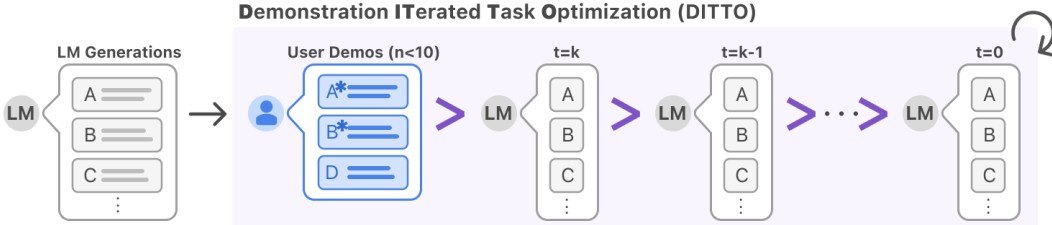

Figure 1: **DITTO iteratively aligns LLMs to demonstrated behavior.** When a user supplies demonstrations (through edits to a model's output, past preferred interaction history, or writing examples from scratch), DITTO treats these demonstrations as preferred to all model behavior, including earlier iterations of the trained model. Using demonstrations as feedback allows for cheap generation of online comparison data and enables few-shot alignment with just a handful of samples.

This paper introduces a framework for aligning LLMs to specific settings by providing a small number of *demonstrations* (Fig. 1). Rather than using prompts, principles, or pairwise preferences, we show that we can achieve strong alignment with individuals by leveraging a small number of user-provided examples of desired behavior. These examples can be drawn from a user's existing interaction logs, or from direct edits made to LLM outputs. Our approach, DITTO, scaffolds a handful of these demonstrations ($< 10$) into a substantial dataset of preference comparisons, by treating users' demonstrations as preferred over model output from both the original LLM and models' earlier training iterations. This augmented dataset of demonstration-grounded comparisons can then be used to update the language model using an alignment algorithm like DPO (Rafailov et al., 2023). We additionally show that DITTO can be interpreted as an online imitation learning algorithm, where data sampled from the LLM is used to distinguish expert behavior. This perspective allows us to prove that DITTO can extrapolate *beyond* the performance of the demonstrator (§3).

Since DITTO focuses on user/task-specific alignment, we benchmark DITTO through (1) an evaluation on datasets of author-specific writing (§4.1) and (2) a user evaluation (§4.2) on real-world tasks defined by human participants. Our author-specific datasets include writing from blog posts to emails to articles. We find that win rates for DITTO outperform methods like SFT (avg. 11% pt. increase), self-play methods like SPIN (20.2% pt.), and few-shot prompting (33.4% pt.) on Mistral 7B. DITTO's advantage holds even when few-shot prompting is done on a more powerful LLM (GPT-4, 18% pt.). Next, we conduct a user study ($N = 16$), asking individuals to edit generations from GPT-4 in an email-writing task. We use finalized demonstrations as inputs for DITTO. In these realistic user evaluations, DITTO's advantage becomes clearer: DITTO continues to outperform baselines, including few-shot prompting (23.9% pt.), user-constructed prompts (27.9% pt.), and SFT (12% pt.). Finally, in a direct comparison between demonstrations and pairwise feedback, we show that using demonstrations with DITTO is an order of magnitude more sample-efficient for individuals than soliciting pairwise preferences.

## 2 RELATED WORK

**LLMs and Preference Finetuning.** Large language models trained on vast amounts of data have been known to perform well with careful prompting (Brown et al., 2020b; Wei et al., 2022). Prompting, however, can be incredibly tedious (Zamfirescu-Pereira et al., 2023) to design and often sensitive to variations. Thus, it has become necessary to either finetune these models on large curated instruction following datasets (Mishra et al., 2022; Thoppilan et al., 2022; Chung et al., 2022) and/or employ RLHF, where the LLM is trained to maximize a reward function learned from human preferences as a contextual bandit (Ziegler et al., 2019). Typically, this is done using policy-gradient style methods (Williams, 1992; Schulman et al., 2017) though more recent works learn directly from preference data (Rafailov et al., 2023; Hejna et al., 2024; Azar et al., 2023). While these methods are effective at tasks like summarization (Stiennon et al., 2020; Wu & Hu, 2018; Wu et al., 2024) and instruction following (Ouyang et al., 2022; Nakano et al., 2021) they require thousands to hundreds of thousands of paired comparisons to obtain a quality estimate of reward. This makes them prohibitively expensive for a wide range of applications, such as training a customized writing assistant or building a domain-specific chatbot. Group Preference Optimization (GPO) (Zhao et al., 2023) takes a promising step towards few-shot alignment of LLMs; however, preference groups must be pre-defined for meta-learning, which requires a large dataset. On the other hand, Gao et al. (2024) uses direct edits to distill

latent preferences into prompt-based principles. In place of principles or pairwise feedback, DITTO directly learns preferences from a set of demonstrations, similar to model editing from canonical examples (Hewitt et al., 2024). Drawing from prior studies on programming by demonstration and end-user programming in HCI (Cypher, 1991; Cypher & Halbert, 1993), our work aims at soliciting feedback at a finer-grained level than binary preferences, principles, or prompts.

**Self-Improvement.** Recent works use iterative sampling to improve LLMs. Aproaches like STaR (Zelikman et al., 2022; 2024; Andukuri et al., 2024) are supervised by verifying the correctness of outputs, while Yuan et al. (2024) and Burns et al. (2023) use (potentially stronger) language models as critics. Unlike these approaches, DITTO does not require external signals besides demonstrations, similar to self-play methods like SPIN (Chen et al., 2024). Unlike SPIN—which uses thousands of demonstrations and is targeted more towards SFT scale datasets—DITTO is designed for fast adaptation in the data-limited setting and thus has a few key distinctions. Namely, DITTO does not update the reference policy and uses intermodel comparisons to combat overfitting. We found these changes to be important to obtain good performance with only a handful of demonstrations. In data-abundant settings, other works have shown that an oracle reward function (Gulcehre et al., 2023) or model (Lee et al., 2023; Song et al., 2024) is sufficient to provide feedback. We consider tasks like personalization, for which there is no abundant data or oracle.

**Online Imitation Learning.** DITTO builds on online imitation learning, which appeals to the long-standing success of learning reward functions from comparisons (Fürnkranz et al., 2012; Akrour et al., 2012). Brown et al. (2019) first showed that with ranked demonstrations, one could improve a policy beyond the demonstrator's performance. Follow-ups used automatic noise injection to remove human rankings (Brown et al., 2020a). Other contemporary approaches to online imitation learning are based on adversarial games between reward and policy players (Ziebart et al., 2008; Ho & Ermon, 2016). In our case, we use a KL-constrained formulation, like Watson et al. (2023). Sikchi et al. (2022) generalizes the adversarial game to a ranking game and thus uses generated comparisons like DITTO. Unlike DITTO, however, these approaches explicitly require learning a reward function and are designed for continuous control—not for LLMs.

## 3 DITTO

While prior work uses thousands of comparisons to align LLMs, DITTO instead uses only a handful of expert demonstrations to alter a model's behavior. This type of cheap, rapid adaptation is enabled by our core insight: that online comparison data can be easily obtained from demonstrations.

### 3.1 NOTATION AND BACKGROUND

A language model can be viewed as policy $\pi(y|x)$ that produces a distribution over completions $y$ to a prompt $x$. In RLHF, our objective is to train an LLM to maximize a reward function $r(x, y)$ that measures the quality of a prompt-completion pair $(x, y)$. Typically, a KL-divergence constraint is added to prevent the updated model from straying too far from a base LM (Ziegler et al., 2019), which we denote as $\pi_{\text{ref}}$. Altogether, RLHF methods optimize the following objective,

$$\mathcal{J}_{\text{KL}}(\pi) = \mathbb{E}_{y \sim \pi(\cdot|x), x \sim p} \left[ r(x, y) - \alpha \log \frac{\pi(y|x)}{\pi_{\text{ref}}(y|x)} \right] \tag{1}$$

which maximizes the expected reward over the prompt distribution $p$ subject to a KL-constraint modulated by $\alpha$. Usually, this objective is optimized using a comparison dataset of the form $\{(x, y^w, y^l)\}$, where the "win" completion $y^w$ is preferred to the "loss" completion $y^l$, which we write as $y^w \succeq y^l$.

While this objective is ubiquitous in prior work (Ouyang et al., 2022; Rafailov et al., 2023), it is typically applied in the context of population-based reward functions learned from large comparison datasets collected via a multitude of annotators. In contrast, we consider $r(x, y)$ to be the objective of a single individual. In this regime, collecting thousands of comparisons from one user is infeasible. Instead, we assume access to a small dataset of expert demonstrations, denoted $\mathcal{D}_E$. We assume these demonstrations to be generated from the expert policy $\pi_E = \arg\max_\pi \mathbb{E}_{y \sim \pi(\cdot|x), x \sim p}[r(x, y)]$, which maximizes reward in expectation. While demonstrations are typically used for SFT, such approaches typically struggle in data-limited settings. On the other hand, it can be difficult to prompt a model to "overcome" the priors induced by its RLHF training. DITTO, as described in the next section, addresses these problems by directly generating comparison data using LM outputs and expert demonstrations. This means that unlike synthetic data generation paradigms (Lee et al., 2023), DITTO does not require a model that performs well at the given task a priori.

## 3.2 DITTO

The key insight of DITTO is that the LM itself, along with the expert demonstrations, can generate comparison datasets for alignment, removing the need to collect a large number of pairwise preferences. This results in a contrastive-like objective, where the expert demonstrations are positives. Here we provide an intuitive explanation of DITTO; later we provide a more theoretical derivation in §3.3.

---

**Algorithm 1: DITTO**

**Input :** LM $\pi_{\text{ref}}$, demos $\mathcal{D}_E = \{(x_i, y_i^E)\}_{i \in N}$, sample size $M$, sample frequency $K$
**Init :** $\pi_0 \leftarrow \textbf{SFT}(\pi_{\text{ref}}, \mathcal{D}_E)$, $t = 0$
**while** *not converged* **do**
    $\mathcal{D}_t \leftarrow \cup_{i=1}^N \{(x_i, y_j \sim \pi_t(\cdot|x_i)\}_{j=1}^M$
    **for** $k = 1, 2, 3, ..., K$ **do**
        Sample batch $B = \{(x, y^w, y^l)\}$ of
        comparisons from induced ranking:
            $\mathcal{D}_E \succeq \mathcal{D}_t \succeq \mathcal{D}_{t-1} \succeq ... \succeq \mathcal{D}_0$
        $\pi_t \leftarrow \text{DPO}(\pi_t, B)$ # Update policy
    $t \leftarrow t + 1$

---

**Generating Comparisons.** Consider a completion sampled from the expert policy, $y^E \sim \pi_E(\cdot|x)$. By virtue of being "expert", $y^E$ is likely to have *high* reward, as $\pi_E$ is definitionally the reward maximizer in expectation. Consequently, we would expect samples from *any* other policy $\pi$ to have rewards less than or equal to those of $\pi_E$, *i.e.*, $\forall \pi, \mathbb{E}_{\pi_E}[r(x, y)] \geq \mathbb{E}_\pi[r(x, y)]$. Using this observation, we can construct comparisons $(x, y^E, y^\pi)$ where $y^E \succeq y^\pi$ by simply sampling completions $y^\pi \sim \pi(\cdot|x)$ for every demonstration-prompt pair in $\mathcal{D}_E$. Though such comparisons are derived from policies instead of individual examples, they have proven effective in prior work (Brown et al., 2020a). A naïve approach for DITTO would then optimize Eq. (1) using this dataset and an off-the-shelf RLHF algorithm. Doing so would increase the probability of the expert responses while decreasing the probability of the current model samples, unlike standard finetuning which only does the former. Crucially, using samples from $\pi$ allows us to construct an unbounded preference dataset given only a few demonstrations. However, we can do better by considering the temporal aspect of the learning process.

**From Comparisons to Rankings.** Using comparisons only between the expert and single policy $\pi$ may be insufficient for obtaining good performance. Doing so decreases likelihoods only at that specific $\pi$, leading to the overfitting problems that plague SFT in low-data regimes. Analogous to replay in RL (Mnih et al., 2015), we can consider data generated from *all* policies learned over time.

At the first iteration, let the initial policy be $\pi_0$. We can sample from this policy to assemble a dataset $\mathcal{D}_0 = \{(x, y^{\pi_0})\}$. Then, we can generate comparison data for RLHF as $y^E \succeq y^{\pi_0}$, which we denote as $\mathcal{D}_E \succeq \mathcal{D}_0$ for brevity. Using these induced comparisons, we update $\pi_0$ to obtain a new policy $\pi_1$. By definition, $\mathbb{E}_{\pi_E}[r(x, y)] \geq \mathbb{E}_{\pi_1}[r(x, y)]$ as well. It follows that we can also generate comparisons using $\pi_1$ as $\mathcal{D}_E \succeq \mathcal{D}_1$. Continuing this procedure, we generate a progressively more diverse comparison dataset using all prior policies. We refer to these as "replay" comparisons.

While this approach is theoretically consistent, it decreases the likelihood of the LM everywhere except at expert demonstrations. Though permissible in data rich scenarios, this may also lead to overfitting with a small $\mathcal{D}_E$. However, if we assume that the policy improves at each iteration, i.e. $\mathbb{E}_{\pi_{t+1}}[r(x, y)] \geq \mathbb{E}_{\pi_t}[r(x, y)]$, then we can also consider comparisons between policies during the course of learning. Unlike comparisons with the expert, we do not guarantee that this holds; in practice, however, we found that models tended to improve with each iteration, perhaps owing to the convexity of both reward modeling and Eq. (1). This lets us sample comparisons between the complete ranking of policies:

$$\mathcal{D}_E \succeq \mathcal{D}_t \succeq \mathcal{D}_{t-1} \succeq ... \succeq \mathcal{D}_1 \succeq \mathcal{D}_0. \tag{2}$$

The effect of adding these "intermodel" and "replay" comparisons is that the likelihoods of earlier samples (e.g., those in $\mathcal{D}_1$) are pushed down more than those of later samples (e.g., those in $\mathcal{D}_t$), smoothing the implicit reward landscape. Our practical implementation aggregates a handful of these intermodel comparisons in addition to comparisons with the expert.

**A Practical Algorithm.** In practice, the DITTO algorithm is an iterative procedure comprised of *three* simple components as outlined in Algorithm 1. *First*, we begin by running supervised fine-tuning on the set of expert demonstrations for a limited number of gradient steps. We set this to be the initial policy $\pi_0$. *Second*, we sample comparisons: at most $K$ times during the training process, we construct a new dataset $\mathcal{D}_t$ by sampling $M$ completions from $\pi_t$ for each of the $N$ demonstrations in $\mathcal{D}_E$ and add it to the ranking over policies Eq. (2). When sampling comparisons from Eq. (2) each

batch $B$ is comprised of 70%"online" comparisons $\mathcal{D}_E \succeq \mathcal{D}_t$, 20% "replay" comparisons of the form $\mathcal{D}_E \succeq \mathcal{D}_{i<t}$, and 10% "intermodel comparisons" of the form $\mathcal{D}_{i \leq t} \succeq \mathcal{D}_{j<i}$. *Finally*, we update the policy using RLHF. Specifically, using batches sampled via the aforementioned procedure, we update the policy $\pi_t$ to obtain $\pi_{t+1}$ using the DPO (Rafailov et al., 2023) loss function

$$\mathcal{L}_{\text{DPO}}(\pi, \mathcal{D}) = -\mathbb{E}_{(x,y^w,y^l)\sim\mathcal{D}} \left[ \log \sigma \left( \alpha \log \frac{\pi(y^w|x)}{\pi_{\text{ref}}(y^w|x)} - \alpha \log \frac{\pi(y^l|x)}{\pi_{\text{ref}}(y^l|x)} \right) \right].$$

where $\sigma$ is the logistic function from the Bradley-Terry preference model. During each update, we do not update the reference model $\pi_{\text{ref}}$ from the SFT policy to avoid straying too far from initialization. DITTO can support any direct preference optimization method as part of the final step (e.g. KTO Ethayarajh et al. (2024) or SimPO Meng et al. (2024)). In practice, we found that the exact choice of the preference optimization algorithm had limited downstream effect, so we defaulted to DPO for all experiments.

### 3.3 DERIVING DITTO AS ONLINE IMITATION LEARNING

DITTO can be derived through an *online imitation learning* perspective, where expert demonstrations are used in conjunction with online data to simultaneously learn a reward function and policy. Specifically, the policy player maximizes expected reward $\max_\pi \mathcal{J}(\pi, r)$, as the reward player minimizes its loss $\min_r \mathcal{L}(\mathcal{D}^\pi, r)$ over an online dataset $\mathcal{D}^\pi$. Concretely, we instantiate this optimization problem using the policy objective in Eq. (1) and the standard reward modeling loss

$$\min_r \left\{ -\mathbb{E}_{(x,y^w,y^l)\sim\mathcal{D}_\pi} \left[ \log \sigma(r(x,y^w) - r(x,y^l)) \right] \text{ s.t. } \pi = \arg \max_\pi \mathcal{J}_{\text{KL}}(\pi, r) \right\}. \tag{3}$$

As done in prior work (Sikchi et al., 2022), we take $\mathcal{D}^\pi$ to be a dataset of comparisons such that $y^\pi \succeq y^{\pi'}$ if $\mathbb{E}_\pi[r(x,y)] \geq \mathbb{E}_{\pi'}[r(x,y)]$. The $\pi$ superscript indicates that $\mathcal{D}^\pi$ contains *online* comparisons between $\pi$ and the expert $\pi_E$. By using different choices of regularizers and comparison data, one can arrive at different inverse RL (IRL) objectives (Ho & Ermon, 2016).

**Deriving DITTO.** The first step in simplifying Eq. (3) is addressing the inner policy maximization. Fortunately, from Ziebart (2010) we know that the policy objective $\mathcal{J}_{\text{KL}}$ has a closed form solution of the form $\pi^\star(y|x) = \pi_{\text{ref}}(y|x)e^{r(x,y)/\alpha}/Z(x)$ where $Z(x)$ is the partition function normalizing the distribution. Notably, this establishes a bijection between policies and reward functions which we can use to eliminate the inner optimization. By rearranging this solution, we can write the reward function $r$ as

$$r(x,y) = \alpha \log \frac{\pi^\star(y|x))}{\pi_{\text{ref}}(y|x)} - \alpha \log Z(x).$$

Furthermore, prior work (Rafailov et al., 2024) shows that this reparameterization can express any reward function. Thus, we can perform a change of variables from $r$ to $\pi$ by substitution into Eq. (3), giving us the DITTO objective

$$\min_\pi -\mathbb{E}_{\mathcal{D}^\pi} \left[ \log \sigma \left( \alpha \log \frac{\pi(y^w|x)}{\pi_{\text{ref}}(y^w|x)} - \alpha \log \frac{\pi(y^l|x)}{\pi_{\text{ref}}(y^l|x)} \right) \right].$$

Note that like DPO, we implicitly estimate the reward function. Unlike DPO, DITTO depends on an *online* dataset of preferences $\mathcal{D}^\pi$. At a minimum, the online preference dataset ought to contain comparisons $\pi_E \succeq \pi, \forall \pi$. However, any preferences consistent with the ground-truth reward function can additionally be used. We leave this exploration to future work.

**Why does DITTO work better than SFT alone?** One reason for DITTO's relatively high performance is that it uses far more data than SFT by generating comparisons. Another is that online imitation learning methods can, in some circumstances, perform *better* than the demonstrator while SFT only mimics the demonstrations. While this is known in the IRL community, we show the following result in Appendix B to relate DITTO's ability to extrapolate beyond the demonstrator to two divergence measures.

**Lemma 3.1.** *(Adapted from Brown et al. (2020a)) Let $\pi^\star$ be the optimal policy for Eq. (1) and $\hat{\pi}$ be the policy estimated by DITTO using expert demonstrations $\mathcal{D}_E$. Extrapolation beyond the demonstrator, i.e. $\mathbb{E}_{\hat{\pi}}[r(x,y)] > \mathbb{E}_{\mathcal{D}_E}[r(x,y)]$ is guaranteed if $\mathcal{J}_{KL}(\pi^\star) - \mathbb{E}_{\mathcal{D}_E}[r(x,y)] > \alpha D_{KL}(\hat{\pi}||\pi^\star) - \alpha D_{KL}(\hat{\pi}||\pi_{ref})$.*

## 4 EXPERIMENTS

We first outline benchmarks, focusing on tasks with subjective preferences (e.g., email writing, essays, articles). We then discuss automatic evaluation, compare DITTO to several baselines, and outline results. Finally, we conduct a user study with DITTO, soliciting demonstrations from participants.

### 4.1 STATIC BENCHMARKS

**Data**    Measuring few-shot alignment with DITTO requires demonstrations from individuals instead of aggregated datasets. We therefore build on prior *Author Attribution* (AA) datasets. The AA task requires one to determine which author $a$ from a set of authors $A$ wrote a specific document. We can reframe prior AA classification tasks as effective alignment: aligning an LLM to a specific author should result in *generations* that are more likely to be attributed to the same author. We collect data from 20 distinct authors from two sources: (1) emails and blog posts from the CMCC dataset (Goldstein et al., 2008) that contain only one author and (2) news articles from the CCAT dataset (Lewis et al., 2004). Our AA benchmarks consist of a diverse range of tasks at the author-level; tasks span from writing financial editorials to opinion pieces on controversial topics. High performance on CMCC / CCAT50 requires non-trivial generalization across prompts. For more dataset details and example tasks, we refer the reader to Appendix C.

**Splits and Preprocessing**    Some of our benchmarks have more writing samples per author than others. While the original CCAT can have more than 50 samples per author, CMCC can have as few as 12. To control for sample count, we randomly select the smallest set of demonstrations available from each author across our training splits (12) for our experiments. We randomly select 10 authors from each dataset, use 7 samples to train, and split the remainder into test and validation. Table 4 in the Appendix describes the finalized train/val/test counts across each benchmark.

**Models and Baselines**    Alongside DITTO, we evaluate supervised fine-tuning (**SFT**), testing if simply fine-tuning on the expert demonstrations $\mathcal{D}_E$ for longer is effective. We also evaluate **SPIN** (Chen et al., 2024), an iterative self-play method designed to replace SFT. Finally, we test **zero-shot** and **few-shot** prompting, including demonstrations directly in the model's context. For few-shot prompting, we add the entire train set of an author's demonstrations in-context. We additionally tried to prompt engineer with zero-shot constraints (e.g., prompting the model to not sound like an LM), with limited success (see Appendix E). Our experiments require an instruction following LLM. We use Mistral Instruct v0.2 7B as a starting point (Jiang et al., 2023) and train using LoRA (Hu et al., 2021). We did try full finetuning for a handful of authors but observed no significant difference. We therefore used LoRA for all experiments. Finally, we compare against zero/few-shot prompting with a more powerful LLM (GPT-4). Hyperparameter details are in Appendix D.

**Automatic Evaluation**    Given that our datasets contain a total of 20 authors, we must train and evaluate a large set of models (20 authors x 7 training paradigms = 140 models). To facilitate the evaluation process, we use GPT-4[2] to compare the outputs of models across various conditions. Prior work has used GPT to both annotate and evaluate text (Zheng et al., 2024). In general, performance lags behind human evaluation; however, for detecting authorship and style similarity, prior work has shown that model-based classification is actually more reliable than non-expert humans (Krishna et al., 2020; Hallinan et al., 2023; Liu et al., 2024; Liu & May, 2024) and GPT-4 eval generally outperforms other automatic metrics (Kim et al., 2023), allowing us to scale hyperparameter search and run evaluation in a more cost-effective manner.

In our setting, we use GPT-4 to determine if a text sounds more or less like a specific author. Given an author-written text $t$ and two pairs of generated text from different conditions $a$ and $b$, we prompt GPT-4 to select the text that most closely matches the validation or test text $t$, and compute averaged head-to-head win rates. To account for ordering bias, we swap orders and average the judgments. Our evaluation prompt and performance benchmarking details are outlined in Appendix F.

**Results**    Our main results, evaluated with GPT-4 eval, are summarized in Table 1. Averaged across all authors, DITTO outperforms all baselines, with an average 77.09% win-rate across both CMCC (71.67%) and CCAT50 (82.50%). On CCAT50, DITTO outperforms all baselines across authors but one. On CMCC, DITTO outperforms all other baselines for 5/10 authors, followed by few-shot

---

[2]We use the `gpt-4-0613` version of GPT-4. We observed that Turbo versions of GPT-4 were more biased towards their own outputs. Queries were run between December 20th, 2023 to May 10th, 2024.

| Data | | Method | $a_{avg}$ | $a_1$ | $a_2$ | $a_3$ | $a_4$ | $a_5$ | $a_6$ | $a_7$ | $a_8$ | $a_9$ | $a_{10}$ |
|------|------|--------|-----------|-------|-------|-------|-------|-------|-------|-------|-------|-------|----------|
| CMCC | GPT | zero-shot | $31.89_{3.05}$ | 43.06 | 29.17 | 22.22 | 37.04 | 18.52 | 42.59 | 19.44 | 40.28 | 40.28 | 31.48 |
| | | few-shot | $63.89_{3.18}$ | **73.61** | 68.06 | 62.50 | 62.04 | 55.56 | 64.81 | **75.93** | **63.89** | 40.28 | 68.52 |
| | Mistral | zero-shot | $27.33_{2.24}$ | 34.72 | 30.56 | 16.67 | 29.63 | 27.78 | 30.56 | 19.44 | 38.89 | 19.44 | 26.85 |
| | | few-shot | $46.89_{4.76}$ | 61.11 | **76.39** | 26.39 | 30.56 | 42.59 | 52.78 | 37.04 | 41.67 | 54.17 | 54.63 |
| | | SPIN | $51.56_{3.85}$ | 56.94 | 48.61 | 56.94 | 40.74 | 73.15 | 48.15 | 59.26 | 59.72 | 31.94 | 38.89 |
| | | SFT | $56.78_{7.04}$ | 18.06 | 27.78 | **86.11** | 74.07 | 58.33 | 43.52 | 64.81 | 47.22 | 81.94 | 58.33 |
| | | DITTO | $\mathbf{71.67_{2.30}}$ | 62.50 | 69.44 | 79.17 | **75.93** | **74.07** | 67.59 | 74.07 | 58.33 | **81.94** | **71.30** |
| CCAT | GPT | zero-shot | $19.35_{1.40}$ | 19.44 | 24.07 | 25.00 | 18.52 | 12.96 | 20.37 | 12.04 | 23.15 | 16.67 | 21.30 |
| | | few-shot | $53.70_{2.19}$ | 64.81 | 53.70 | 61.11 | 53.70 | 47.22 | 44.44 | 45.37 | 61.11 | 52.78 | 52.78 |
| | Mistral | zero-shot | $18.06_{1.61}$ | 13.89 | 23.15 | 15.74 | 12.96 | 13.89 | 22.22 | 17.59 | 14.81 | 28.70 | 17.59 |
| | | few-shot | $40.37_{2.33}$ | 56.48 | 45.37 | 35.19 | 32.41 | 41.67 | 39.81 | 46.30 | 35.19 | 34.26 | 37.04 |
| | | SPIN | $62.13_{3.11}$ | 56.48 | 69.44 | 55.56 | **82.41** | 70.37 | 54.63 | 58.33 | 54.63 | 51.85 | 67.59 |
| | | SFT | $73.89_{2.50}$ | 61.11 | 62.04 | 76.85 | 72.22 | 80.56 | 81.48 | 80.56 | 68.52 | 82.41 | 73.15 |
| | | DITTO | $\mathbf{82.50_{1.93}}$ | **77.78** | **72.22** | **80.56** | 77.78 | **83.33** | **87.04** | **89.81** | **92.59** | **83.33** | **80.56** |

Table 1: **GPT-4 Eval**: Head-to-head win rates between methods across benchmark test splits. DITTO outperforms all baseline methods on average and across a plurality of individual authors. $a_1...a_{10}$ represents a single model trained on one of ten sampled authors from each dataset (see §4). Results are averaged across 3 runs, with 3 samples generated from each model with temperature 1.0. We also report win rates averaged across authors, along with standard error of the mean ($avg_{sem}$).

prompting for 3/10. While SFT serves as a strong baseline (56.78% on CMCC, 73.89% on CCAT), DITTO provides an average ↑11.7% pt. win rate improvement compared to SFT alone.

Prompted baselines also lag far behind DITTO, especially zero-shot (including closed-source) models (avg. ↓54.4% pt. decrease on Mistral, ↓51.5% pt. on GPT-4). While zero-shot GPT-4 is already finetuned using RLHF, we suspect that this training feedback differs significantly from that of authors in both CMCC and CCAT50. Adding all train instances as a few-shot prompt does help: win rates for few-shot prompting increase compared to zero-shot for both Mistral (↑20.94% pt.) and GPT-4 (↑22.95% pt.) based LLMs. However, including few-shot examples still falls behind applying DITTO (avg. ↓37.35% pt. decrease for Mistral; ↓26.99% pt. for GPT-4). Varying the number of demonstrations in the few-shot prompt also yields no improvement (Fig. F.2 in Appendix). We suspect the underlying RLHF priors for out-of-the-box LLMs are fairly strong. Qualitatively, few-shot generations still sound GPT-generated relative to DITTO (Table 7 in Appendix).

While we do test another self-improvement training method (SPIN), we find that performance is lower than DITTO (avg.↓ 9.3% pt.)—we suspect that design decisions for SPIN (e.g., updating the reference policy, excluding interpolicy / replay comparisons) are targeted towards SFT-scale datasets. We ablate these decisions in §5.1 and propose reasons for performance degradation.

Finally, we ran an ANOVA test to determine whether there were significant differences between conditions, and then ran a Tukey test to identify which specific conditions were significant. DITTO's improvements are significant ($p < 0.05$) compared to all other conditions in Table 1, excluding few-shot GPT on CMCC.

## 4.2 USER STUDY: TESTING GENERALIZATION TO NATURALISTIC TASKS

Our static benchmarks have focused on pre-existing author attribution datasets, using GPT-4 to measure alignment. However, GPT-4 eval exhibits a self-enhancement bias, likely inflating performance for LLM-like generations (Zheng et al., 2024; Panickssery et al., 2024). We therefore evaluate DITTO in a *more naturalistic* setting; we conduct a user study to evaluate DITTO and ask users to provide demonstrations for a range of tasks. As baselines, we use zero-shot and few-shot prompted GPT-4, along with SFT. Additionally, we ask participants to self-prompt models by iteratively authoring their own prompts to steer the model outputs. Zero-shot, few-shot, and self-prompt emulate what most users would do today to steer LLMs, and SFT provides a strong finetuning baseline.

We recruit 16 participants from social media (Twitter). Many of our participants were Ph.D. students familiar with prompting LLMs; therefore, our self-prompt baseline offers a strong baseline for additional prompt engineering. Participants were paid $30 / hr; our study was approved by an IRB.

**User Study Outline** The user study consists of two parts. In the first part, we ask participants to specify four email-writing tasks (e.g., *Write an email to your advisor asking for feedback*). Participants are asked to provide two demonstrations for two of the tasks (4 training demonstrations in total). To

help brainstorm tasks, we generate concrete task suggestions with GPT-4; participants could select from among these or provide their own custom tasks. We randomly split two task prompts into train, and saved two for testing; participants gave two demonstrations each for both the training prompts, to mimic a user willing to only put in minimal effort. Users were provided with default generations from GPT-4 to aid authoring demonstrations, which they could edit or ignore. In the second part, we use the two tasks from the test set and show participants generations across all methods. We sampled one output from each method (self-prompt, zero-shot, few-shot, SFT, and DITTO), and solicited 10 pairwise preferences for each test prompt (resulting in 20 preferences total for each user). In all, we collect a total of 320 pairwise preferences across 16 users. All comparisons are done blinded to the condition. Additional user study details (e.g., interface, examples of demonstrated feedback, prompts for generating tasks, etc.) are in Appendix G.

**Results**  Our user study results corroborate findings from static benchmarks. DITTO outperforms baseline methods in aligning to demonstrated preferences (Table 2), with DITTO (68.8% win-rate) > SFT (55.5%) > few-shot (51.6%) > self-prompt (46.9%) > zero-shot (27.3%). DITTO is significantly better than all other methods (ANOVA + Tukey test, $p < 0.05$). Additionally, users generally struggle with verbalizing preferences into prompts: self-prompting slightly underperforms providing demonstrations in a few-shot prompt, and substantially underperforms DITTO. We also qualitatively observe that users often edit nearly half of the default output from GPT-4 when authoring demonstrations (examples in Appendix G), with average normalized Levenshtein edit distance = 0.43. Large edits to the output alone highlight the effectiveness of demonstrated feedback as an interaction.

| Method | | Win Rate |
|---|---|---|
| GPT-4 | zero-shot | 27.3 |
| | few-shot | 51.6 |
| | self-prompt | 46.9 |
| SFT | | 55.5 |
| DITTO | | **68.8** |

Table 2: **User Study Results.** In head-to-head human annotated win rates, DITTO outperforms self-prompted, few-shot, and zero-shot GPT-4 baselines, along with SFT.

To better understand why users in our study selected DITTO outputs, we fit a Fightin'-Words model to identify lexical differences between generations from GPT-4 and DITTO (Monroe et al., 2008). Fightin'-Words is used to identify words that are statistically significantly different in frequency between corpora. It generates log-odds ratios and z-scores, which measure how likely a word is to appear in one corpus compared to another. Many of the words that appear in GPT generated outputs compared to DITTO (Table 6) come from cliche phrases: "greatly appreciate your time and understanding" or "hope this message finds you well." Compared to DITTO, GPT also regularly generates sentences mentioning "trust" and "initiative" in emails drafted to close collaborators. We suspect GPT's writing style is tightly coupled to its RLHF priors.

## 5 WHEN DOES DITTO WORK?

A user must decide on several prerequisites before using DITTO, from how many demos they have to how many negatives they must sample from the LM. We explore the impact of these decisions and focus on CMCC, as it covers a broader range of tasks than CCAT. We additionally analyze sample efficiency of demos vs. pairwise feedback in our user study.

| Ablation | Win Rate |
|---|---|
| Sample only at start | 57.3 |
| DITTO | **70.1** |
| $\rightarrow$ remove interpolicy | 68.1 |
| $\rightarrow$ remove replay | 63.6 |
| $\rightarrow$ update $\pi_{\text{ref}}$ | 45.8 |

Table 3: **Head-to-head win rates across DITTO algorithm ablations on CMCC.** We experiment with **sampling** all negatives **at the start**, ablating **replay** and **interpolicy** comparisons, and updating the reference policy.

### 5.1 ALGORITHM PERTURBATIONS

DITTO consists of several hyperparameters: namely, the **number of DITTO iterations** $N = \{1..4\}$ and **negative samples** $M = \{2...10\}$ generated from our sequence of policies. Separately, we ablate components of DITTO, like the use of interpolicy ($\mathcal{D}_{i \leq t} \succeq \mathcal{D}_{j < i}$) and replay ($\mathcal{D}_E \succeq \mathcal{D}_{i < t}$) comparisons. We also test an ablation where we do not re-sample data during training and instead **sample** all negatives **only at the start**, and where we **update** $\pi_{\text{ref}} = \pi_t$ at each iteration like SPIN (Chen et al., 2024). Note that DITTO performance varies from user to user. To account for variance between author-level win rates, we convert each author's averaged win rate to the % improvement from their initial ablation's win rate (e.g., % improvement from 1 DITTO iteration, 2 generated negatives, or 1 training demonstration).

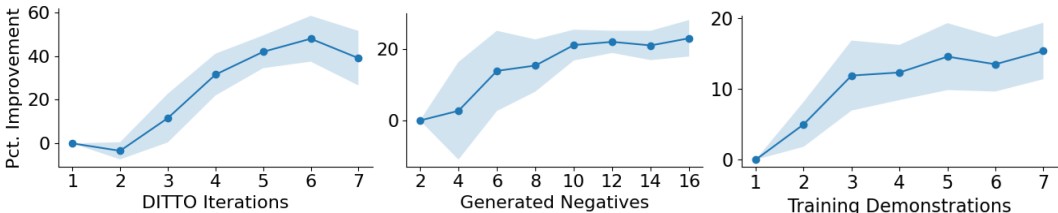

Figure 2: **Head-to-head win rates across DITTO hyperparameter perturbations on CMCC**. First, increasing the number of DITTO iterations improves GPT-4 eval performance (left). Increasing the number of generated negatives also reduces DITTO variance across users while improving DITTO performance (middle). Finally, increasing demos also improves performance, but we observe diminishing returns (right). Error bars correspond to standard error of the mean across authors.

Increasing the number of DITTO iterations generally improves performance (Fig. 2). Comparing Iteration 1 to Iteration 4, we observe a relative 31.5% increase in GPT-4 eval win rates. Improvement is non-monotonic—in Iteration 2, performance drops slightly (-3.4%). Early iterations might yield noisier samples, potentially reducing performance. On the other hand, increasing negative samples monotonically improves DITTO performance. Generating 10 negatives for each demonstration in the training set, for example, yields an 21.09% win-rate improvement compared to just 2. Furthermore, as we sample more negatives increases, variance in DITTO performance decreases. However, there is a tradeoff associated with increasing the number of negative samples: runtime of DITTO will also increase. In addition, we find that added iterations ($> 6$) eventually result in performance degradation, likely due to overfitting. Sampling more negatives ($> 10$) also yields plateauing performance.

We also find that ablating components of DITTO results in reduced performance (Table 3). If we sample all negatives at the start—instead of iteratively resampling in an online fashion—we observe that win rates compared to using DITTO drop from 70.1% to 57.3%. While iteratively re-sampling improves performance, continuously updating $\pi_{\text{ref}}$ during this online process can significantly degrade performance: win rates drop from 70.1% to 45.8%. We suspect updating $\pi_{\text{ref}}$ results in potential overfitting. Finally, both replay and inter-policy comparisons help DITTO. Removing replay and interpolicy comparisons reduces win rates from DITTO by 6.5 and 2 points respectively.

One potential confound for DITTO's performance is that the instruction-following prior is too strong. Few-shot prompting in particular cannot undo the style trained into an instruction-following LLM. To isolate this effect, we additionally compared DITTO to a fine-tuned and few-shot prompted base variant of Mistral. Compared to DITTO, we still see significant degredations—moreso than the instruction following model. Win rates against DITTO are 9.4 and 10.4 for few-shot and SFT respectively. We suspect that general instruction-following capabilities are required as a "starting point." Jointly learning instruction-following and demonstrated feedback is too difficult a task to learn from a handful of demonstrations.

## 5.2 SAMPLE EFFICIENCY

A key affordance of DITTO is its sample efficiency. In §4, we examined DITTO's performance on the full set of 7 demonstrations from each author. In practice, a user may only provide one or two demonstrations. Therefore, we evaluate sample efficiency across DITTO trained smaller subsets of the full training set $N = \{1...7\}$. Like with our algorithm perturbations, we report per-user normalized win rates (Figure 2). First, we observe that DITTO win rates increase rapidly at the start. From $1 \leq N \leq 3$, normalized performance roughly doubles for each additional demonstration ($0\% \rightarrow 5\% \rightarrow 11.9\%$). However, we observe diminishing returns when supplying extra demonstrations ($4 \leq N \leq 7$, $11.9\% \rightarrow 15.39\%$): performance saturates as demonstrations increase. A key design decision in using DITTO lies in the selection of demonstrations; we additionally suspect that the quality of provided demonstrations likely also affects DITTO performance. We conduct a preliminary analysis of demonstration *cohesiveness* on downstream performance, testing how demonstration similarity affects DITTO performance (Appendix H.2). We additionally revisit this in future work.

## 5.3 HOW DO PAIRWISE PREFERENCES COMPARE AGAINST DEMONSTRATIONS?

A core assumption of DITTO lies in sample efficiency coming from demonstrations. In theory, a user *could* achieve similar performance by labeling many pairwise preferences with

an ideal set of demonstrations in mind. As a preliminary approximation, one author provided demonstrations for the user study and also annotated 500 preference pairs using outputs sampled from the instruction following Mistral 7B (demonstrations in Appendix G.4).

Altogether, we constructed a pairwise preferences dataset $D_{pref} = \{(x, y^i, y^j)\}$, where $y_i \succ y_j$. We then computed win rates between 20 pairs sampled from Mistral trained on (a) 4 demonstrations with DITTO, and (b) on $\{0...500\}$ preference pairs with just DPO. When we sample pairwise preferences from $\pi_{\text{ref}}$ alone, we observe that generated pairs are out-of-distribution relative to the demonstrations—pairwise preferences do not reach a user's demonstrated behavior (results in Fig. 3: "Base policy," in blue). Even when we finetune $\pi_{\text{ref}}$ on the user's demonstrations, we still need $> 500$ preferences to match DITTO performance (Fig. 3: "Demo-finetuned policy," in orange). This is especially damning for methods that align LLMs using samples generated from $\pi_{\text{ref}}$ alone (e.g. Constitutional AI)—preferences generated over OOD samples (relative to the user's true reward) are essentially irrelevant.

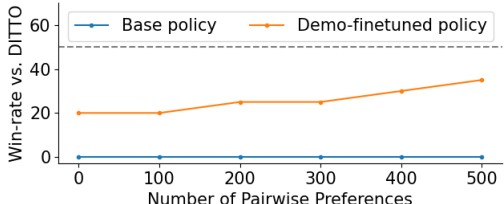

Figure 3: **Demonstrations are more sample efficient than pairwise preferences** for an individual user. We compared DITTO with 4 demos to pairwise prefs sampled from (1) base instruction-following LM $\pi_{\text{ref}}$ and (2) $\pi_{\text{ref}}$ fine-tuned on demos. Applying DPO on 500 pairwise preferences—with samples from $\pi_{\text{ref}}$—yields no improvement compared to DITTO. Even if demos are used to fine-tune $\pi_{\text{ref}}$ before sampling, one must collect many pairwise preferences to approach DITTO.

## 6 CONCLUSION

Current modes for soliciting feedback—like principles or pairwise annotations—cater to population-level preferences. In this work, we instead highlight the effectiveness of using demonstrations as feedback, and show that a limited number of demonstrated behaviors can provide a strong signal for preferences specific to an individual. We also introduce a new technique, DITTO, that cheaply generates online comparison data from demonstrations, and test DITTO's effectiveness across static benchmarks and a user study. Focusing feedback collection at the demonstration level may offer a more diverse overview of individual preferences, and encourage a re-evaluation of the interfaces and interactions used to collect human feedback.

**Limitations** One limitation involves DITTO speed: DITTO is slower than training-free approaches (prompting) and SFT (15 minutes with DITTO vs. 2 minutes with SFT on 7 demonstrations). A bottleneck lies in sampling, though we suspect a mix of prior (e.g., vLLM Kwon et al. (2023)) and future work in LLM inference optimization can improve DITTO's speed. DITTO-ed models also tend to "forget" more general capabilities. For example, we observed that models often refused to write programs (generating "I have no idea how to write code."). In Appendix H.1, we evaluated forgetting on coding tasks with HumanEval Chen et al. (2021), observing some degradation. However, we entirely mitigate all degradations by selectively dropping DITTO's LoRA adapter, and routing instructions between the general instruction-following model and the specialized DITTO LoRA adapter (ala MoE). Finally, because of evaluation and computational constraints, we do not test across model families or sizes. Exploring how DITTO scales is an avenue for future work.

**Future Work** One avenue involves analyzing tradeoffs between types of preference data (e.g., demonstrations vs. preferences vs. principles). While we propose demonstrations as a feedback modality, each type of feedback requires different levels of effort, and the effectiveness depends on the user providing feedback. In addition, preferences provided through demonstrations are often *local* in quality—users provide demonstrated preference in the context of specific domains. Understanding how scaling the amount of *local* demonstrated feedback affects general-purpose model behavior is an avenue for future work. Another key design decision in using DITTO lies in the selection of demonstrations; we additionally suspect that the quality of provided demonstrations likely also affects DITTO performance. While we explore the effect of demonstration cohesiveness in Appendix H.2 (how does demonstration similarity affect DITTO performance), understanding how to select an optimal set of demonstrations for DITTO from a user is an avenue for future work. Given the ability to align models with a handful of demonstrations, DITTO could support new interactions for individual end users to orchestrate many task-specific models curated to their needs; or may motivate inference-only alignment methods for black box LLMs that do not require fine-tuning.

## ETHICS STATEMENT

Demonstrated feedback is a double-edged sword. While DITTO can enable effective personalization of language models, we also suspect that DITTO will be especially useful for model *un*-alignment, amongst a range of other risks (Kirk et al., 2023). However, the current status quo of language model alignment lies with large corporations that practice limited transparency. Models like GPT-4 already espouse dangerous positive stereotypes or unfairly benefit privileged groups due to representation issues in the feedback collection process (Cheng et al., 2023; Ryan et al., 2024).

## ACKNOWLEDGEMENTS

We thank Eric Zelikman, Matt Jörke, Jan-Philipp Fränken, Michael Y. Li, Michael Ryan, Will Held, Shan Rizvi, Suvir Mirchandani, and Jensen Gao for helpful discussions and feedback. We also thank members of the SALT Lab and the Stanford HCI / NLP groups.

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

# A APPENDIX

## B   DERIVING DITTO AS ONLINE IMITATION LEARNING

For understanding the provided derivations, it is helpful to be familiar with the fixed point solution for Eq. (1), which was first derived for maximum entropy RL (Ziebart, 2010).

$$Q^*(x,y) = r(x,y) \quad \text{(because contextual bandit)}$$

$$V^*(x) = \alpha \log \mathbb{E}_{y \sim \pi_{\text{ref}}(\cdot|x)} \left[ e^{r(x,y)/\alpha} \right]$$

$$\pi^*(y|x) = \pi_{\text{ref}}(y|x) e^{(r(x,y) - V^*(x))/\alpha} = \frac{1}{Z(x)} \pi_{\text{ref}}(y|x) e^{r(x,y)/\alpha}$$

where $Z(x) = e^{V^*(x)/\alpha} = \mathbb{E}_{y \sim \pi_{\text{ref}}(\cdot|x)} \left[ e^{r(x,y)/\alpha} \right]$. Using this information, in conjunction with Equation 1, we can a number of useful inequalities between $\pi^*$, $\pi_{\text{ref}}$, and an arbitrary $\pi$.

### B.1   DERIVING DITTO

Here we provide a more detailed derivation of DITTO from an online imtiation learning perspective. In particular, we consider the common two-player min-max interpretation of imitation learning (Ziebart et al., 2008; Sikchi et al., 2022), but do so with general objective functions.

$$\min_r \mathcal{L}(\mathcal{D}^\pi, r) \quad \max_\pi \mathcal{J}(\pi, r)$$

In this formulation, $\mathcal{D}^\pi$ is a dataset of preferences such that $y^\pi \succeq y^{\pi'}|x$ if $\mathbb{E}_\pi[r(x,y)] \geq \mathbb{E}_{\pi'}[r(x,y)]$, i.e. one completion is preferred to another if the corresponding policy has higher expected reward. This framework generalizes prior work. For example, we limit ourselves to only comparing the expert policy $\pi_E$ to the current policy $\pi$, and add a regularizer, we can obtain the maximum entropy IRL objective from Ho & Ermon (2016). Choosing $\mathcal{J}_{\text{KL}}$ as the policy objective function and maximum likelihood on the Bradley-Terry model as the reward objective we get the following optimization:

$$\min_r -\mathbb{E}_{(x,y^w,y^l) \sim \mathcal{D}_\pi} \left[ \log \sigma(r(x,y^w) - r(x,y^l)) \right], \quad \max_\pi \mathcal{J}_{\text{KL}}(\pi, r)$$

where $\mathcal{J}_{\text{KL}}$ is the KL-constrained RL objectve from before, but now dependent on the learned reward function. We then select an ordering for the optimization, by making policy learning the "inner" objective as done in Ho & Ermon (2016). Sikchi et al. (2022) makes connections between this choice and game theory. This results in the same equation in the main paper, repeated here for clarity.

$$\min_r \left\{ -\mathbb{E}_{(x,y^w,y^l) \sim \mathcal{D}_\pi} \left[ \log \sigma(r(x,y^w) - r(x,y^l)) \right] \text{ s.t. } \pi = \arg\max_\pi \mathcal{J}_{\text{KL}}(\pi, r) \right\},$$

We can then re-arrange the fixed point equations from maximum entropy RL, obtaining the "DPO-trick":

$$r(x,y) = \alpha \log \frac{\pi^*(y|x))}{\pi_{\text{ref}}(y|x)} - \alpha \log Z(x).$$

This alone, however, is insufficient to obtain a representation for the optimal policy as naively substituting the above does not garuntee that the domain of reward functions can be fully expressed by such a reparameterization in terms of the policy. Fortunately, prior work have established both that such a reparatermization is equally expressive (Watson et al., 2023; Ng et al., 1999) and that it does not affect the preference model Hejna et al. (2024); Rafailov et al. (2024). Completing this substitution yields the main DITTO objective.

However, DITTO is compatible with other algorithms, such as traditional RL methods, so long as they can be used to solve for the KL-constrained RL objective in Eq. (1). Instead of using the DPO trick, one could use a few steps of a policy gradient algorithm to update the policy.

**Distributional versus Point-wise Preferences.** One thing to note is that we construct preferences for DITTO from distributional preferences, ie $\mathbb{E}_{\pi_1}[r(x,y)] \geq \mathbb{E}_{\pi_2}[r(x,y)]$. However, this only guarantees that completions from one policy are preferred to another in expectation, not necessarily that every realized preference pair follows this relationship. We found that his choice works well in practice, and is actually common in prior work. For example, Brown et al. (2020a) uses a sequence of policies ranked by expected return in combination with a Bradley-Terry model. Appendix C of

Stephan et al. (2024) shows that artificially sampling comparisons between two policies is consistent with a Bradley-Terry reward model. Another possible view of this is that DITTO ends up optimizing an upper bound on the standard reward modeling loss:

$$\mathbb{E}_{\pi^w, \pi^l \sim \mathcal{D}^\pi}[-\log \sigma(\mathbb{E}_{y \sim \pi^w}[r(x,y)] - \mathbb{E}_{y \sim \pi^l}[r(x,y)])] \leq \mathbb{E}_{\pi^w, \pi^l \sim \mathcal{D}^\pi}[-\log \sigma(r(x,y^w) - r(x,y^l))]$$

which arises from applying Jensen's inequality on the negative log-sigmoid function.

## B.2 ONLINE IMITATION CAN PERFORM BETTER THAN SFT

Here we show that, under some circumstances, online imitation learning is theoretically able to perform better than SFT on the expert dataset. To do this, we require a few building blocks.

**Proposition B.1.** *The objective value $\mathcal{J}_{KL}$ of any policy $\pi$ can be expressed in terms of the optimal policy $\pi^*$ as $\mathcal{J}_{KL}(\pi) = \mathcal{J}_{KL}(\pi^*) - \alpha \mathbb{E}_{x \sim p}[D_{KL}(\pi(\cdot|x)||\pi^*(\cdot|x))]$*

*Proof.* Note that at convergence, the optimal policy obeys the equality $\pi^*(y|x) = \pi_{\text{ref}}(y|x)e^{(r(x,y)-V^*(x))/\alpha}$. Thus, we can rewrite the reward function in terms of the optimal policy as

$$r(x,y) = \alpha \log \frac{\pi^*(y|x)}{\pi_{\text{ref}}(y|x)} + V^*(x)$$

and substitute it into the objective function for the reward.

$$\begin{aligned}
\mathcal{J}(\pi) &= \mathbb{E}_{y \sim \pi(\cdot|x), x \sim p}\left[r(x,y) - \alpha \log \frac{\pi(y|x)}{\pi_{\text{ref}}(y|x)}\right] \\
&= \mathbb{E}_{y \sim \pi(\cdot|x), x \sim p}\left[\alpha \log \frac{\pi^*(y|x)}{\pi_{\text{ref}}(y|x)} + V^*(x) - \alpha \log \frac{\pi(y|x)}{\pi_{\text{ref}}(y|x)}\right] \\
&= \mathbb{E}_{y \sim \pi(\cdot|x), x \sim p}\left[\alpha \log \frac{\pi^*(y|x)}{\pi(y|x)} + V^*(x)\right] \\
&= \mathbb{E}_{x \sim p}[V^*(x)] - \mathbb{E}_{y \sim \pi(\cdot|x), x \sim p}\left[\alpha \log \frac{\pi(y|x)}{\pi^*(y|x)}\right] \\
&= \mathcal{J}(\pi^*) - \alpha \mathbb{E}_{x \sim p}[D_{\text{KL}}(\pi(\cdot|x)||\pi^*(\cdot|x))]
\end{aligned}$$

This also implies that $\pi^*$ is unique (though this is known to be true of MaxEnt RL objectives). This means that provided the reference policy is not already optimal, DITTO is able to improve it.

**Corollary B.2.** *Given $\pi_{ref} \neq \pi^*$, then $\mathcal{J}(\pi^*) > \mathcal{J}(\pi_{ref})$.*

This follows by considering proposition 1 in conjunction with the fact that $\mathcal{J}(\pi^*) \geq \mathcal{J}(\pi_{\text{ref}})$ and the KL-divergence is only zero if both distributions are equal.

**Lemma B.3.** *(Adapted from Theorem 1 of Brown et al. (2020a)) Let $\pi^*$ be the optimal policy for Eq. (1) and $\hat{\pi}$ be the policy estimated by DITTO using expert demonstrations $\mathcal{D}_E$. Extrapolation beyond the demonstrator, i.e. $\mathbb{E}_{y \sim \hat{\pi}(\cdot|x), x \sim p}[r(x,y)] > \mathbb{E}_{x, y \sim \mathcal{D}_E}[r(x,y)]$ is guaranteed if*

$$\mathcal{J}_{KL}(\pi^*) - \mathbb{E}_{\mathcal{D}_E}[r(x,y)] > \alpha \mathbb{E}_{x \sim p}[D_{KL}(\hat{\pi}(\cdot|x)||\pi^*(\cdot|x))] - \alpha \mathbb{E}_{x \sim p}[D_{KL}(\hat{\pi}(\cdot|x)||\pi_{ref}(\cdot|x))].$$

*Proof.* This can be shown via simple sequence of inequalities and application of proposition 1. For brevity, we will omit the expectations over the prompt distribution. We proceed directly.

$$\begin{aligned}
\mathbb{E}_{\hat{\pi}}[r(x,y)] &> \mathbb{E}_{\mathcal{D}_E}[r(x,y)] \\
\mathcal{J}_{\text{KL}}(\hat{\pi}) &> \mathbb{E}_{\mathcal{D}_E}[r(x,y)] - \alpha D_{\text{KL}}(\hat{\pi}||\pi_{\text{ref}}) \\
\mathcal{J}_{\text{KL}}(\pi^*) - \mathcal{J}_{\text{KL}}(\pi^*) + \mathcal{J}_{\text{KL}}(\hat{\pi}) &> \mathbb{E}_{\mathcal{D}_E}[r(x,y)] - \alpha D_{\text{KL}}(\hat{\pi}||\pi_{\text{ref}}) \\
\mathcal{J}_{\text{KL}}(\pi^*) - \alpha D_{\text{KL}}(\hat{\pi}||\pi^*) &> \mathbb{E}_{\mathcal{D}_E}[r(x,y)] - \alpha D_{\text{KL}}(\hat{\pi}||\pi_{\text{ref}}) \\
\mathcal{J}_{\text{KL}}(\pi^*) - \mathbb{E}_{\mathcal{D}_E}[r(x,y)] &> \alpha D_{\text{KL}}(\hat{\pi}||\pi^*) - \alpha D_{\text{KL}}(\hat{\pi}||\pi_{\text{ref}})
\end{aligned}$$

If one wants to directly compare expected rewards, the $-\alpha D_{\text{KL}}(\pi^*||\pi_{\text{ref}})$ term in $\mathcal{J}_{\text{KL}}(\pi^*)$ can simply be moved to the right hand side of the inequality. In practice, we choose a fairly small value of $\alpha$. This means that if the objective value of our optimal policy (reward minus KL) is higher than the average reward of the dataset, then we expect to do better than the demonstrator when our learned policy is closer to the optimal one than the reference.

| Source | Author | Train / Author | Val / Author | Test / Author |
|--------|--------|----------------|--------------|---------------|
| CMCC   | 10     | 7              | 2-3          | 2-3           |
| CCAT   | 10     | 7              | 3            | 3             |

Table 4: **Final Aggregate Benchmark Statistics**

## C   DATASET DETAILS

In all, we collect data from a total of 20 distinct authors from two sources: (1) **CMCC** consists of texts written by 21 students in six different genres (email, essay, interview transcript, blog article, chat, or discussion transcript) covering six different controversial topics (Goldstein et al., 2008). We filter this corpus to include only emails and blog posts, excluding sources where multiple individuals were involved (e.g., chat). (2) **CCAT** (Lewis et al., 2004) consists of articles from Canadian Broadcasting Corporation's French Service, sourced from RCV1-v2 Reuters Corpus dataset. Due to the large number of training paradigms evaluated in this work, we sample articles from 10 authors from each dataset (260 documents total). Table 4 highlights raw counts for each author.

### C.1   TRAIN / TEST GENERALIZATION

Within-author demonstrations across our both our user study and benchmarks span a diverse range of tasks, like opinion pieces, blog posts, recipe writing, requests to meet, etc. Performing well on these benchmarks requires non-trivial generalization. Here, we select a handful of train-test prompts that are representative of the train-test generalization expected from DITTO-ed models.

1. Train: Discuss a recent movie or TV show you watched.
   Test: Share a new recipe you tried and loved.

2. Train: The city of Denver has decided to legalize small amounts of marijuana for persons over 21. How do you feel about this?
   Test: Do you feel the Catholic Church needs to change its ways to adapt to life in the 21st Century?

3. Train: Write an email to your professor seeking advice on research topics for an upcoming project.
   Test: Outline an agenda for a project meeting with a new collaborator.

4. Train: Share personal writing rituals and habits for inspiration.
   Test: Highlight a fellow writer's work and encourage support within the community.

## D   HYPERPARAMETERS AND TRAINING DETAILS

We run a random hyperparameter sweep over a single, randomly selected author from each corpus, using lr = $\{1e-4, 3e-4, 1e-5, 3e-5, 1e-6, 3e-6\}$, epoch = $\{10, 15, 20, 25, 30\}$, and $\beta = \{0.01, 0.05, 0.1\}$. We additionally tune how frequently DITTO samples negatives ($K = \{1, 5, 10\}$); and how many negatives DITTO samples ($M = \{1, 5, 10\}$). Finally, we tuned the replay / expert / intermodel fractions, selecting between 0.2 / 0.7 / 0.1, 0.25 / 0.5 / 0.25 and 0.1 / 0.7 / 0.2. We fixed optimal hyperparameters for each benchmark across all our remaining evaluations. We select hyperparameters from searches conducted on the validation set. All training was conducted on 1 A100 80GB GPU. We use the cosine scheduler for the SFT step, with a warmup ratio of 0.1; and the constant_with_warmup scheduler for DPO with a warmup ratio of 0.25. For a dataset, we train with SFT until BCE train loss on a given batch approaches 1.00 (early stopping); ideally, we want an LLM to not overfit entirely to demos before DPO. Finally, we use AdamW across all experiments.

| Dataset | CMCC | CCAT |
|---|---|---|
| LoRA Rank | 16 | 16 |
| Alpha | 32 | 32 |
| SFT Batch Size | 4 | 4 |
| Learning Rate | 3e-5 | 3e-5 |
| DPO Batch Size | $\approx 24$ | $\approx 24$ |
| DPO Learning Rate | 1e-6 | 1e-6 |
| DPO Grad Steps | 40 | 40 |
| DPO $\beta$ | 0.05 | 0.05 |
| DITTO Negative Samples | 10 | 10 |
| Resample Step-Rate | 10 | 10 |
| Resample Temperature | 1.0 | 1.0 |
| Frac Replay | 0.2 | 0.2 |
| Frac Expert | 0.7 | 0.7 |
| Frac Inter-model | 0.1 | 0.1 |

Table 5: Hyperparameters across benchmark datasets.

## E  FEW-SHOT PROMPT

```
Below are a few writing samples.

### EXAMPLE 1
{prompt_1}

{output_1}

....

### EXAMPLE N
{prompt_N}

{output_N}

Respond to the following prompt in the same way as the writing samples.  Do not
generate output that is GPT-like:
{prompt}
```

Figure 4: Few-shot prompt used to generate outputs for few-shot examples. We additionally test ablations in red text, but find that this reduces win rates for few-shot methods by 4% pts.

## F  GPT-EVAL

### F.1  GPT-EVAL PROMPT

We outline our final evaluation prompt below. We re-prompted for every pair of conditions, swapped generation orders to account for positional bias, and computed an averaged win rate. We sample with temperature = 0.0 for eval, and use GPT-4 0613.

### F.2  GPT-EVAL BENCHMARKING RESULTS

We benchmark the performance of our evaluation setup with human data using CMCC by pairing an author's original text with another author's text. CMCC is a more suitable dataset than CCAT for this evaluation: we can pair texts from different authors that discuss the same topic. In CMCC, there are authors that wrote essays or emails for the same prompt while there are none of such cases in CCAT. For each author in CMCC, we create 50 samples with the aforementioned setup, leading to a total of 950 comparisons. With this setup and using bootstrap sampling, GPT-4 Eval achieves $81.79 \pm 2.42\%$ accuracy.

```
System:   You are an impartial evaluator.

You are an impartial evaluator.  Below is a sample of a human author's writing and two
options.

### HUMAN AUTHOR'S WRITING:
{demo}

### OUTPUT A:
{text_a}

### OUTPUT B:
{text_b}

### Task
Which option was written by the human author based on similarity to the HUMAN AUTHOR'S
WRITING above?  Respond only with a JSON of the following format:
{
   "answer":  "<The option most similar to the HUMAN AUTHOR'S WRITING; either A or B>"
}

ALWAYS REMAIN IMPARTIAL WHEN EVALUATING OUTPUTS.
```

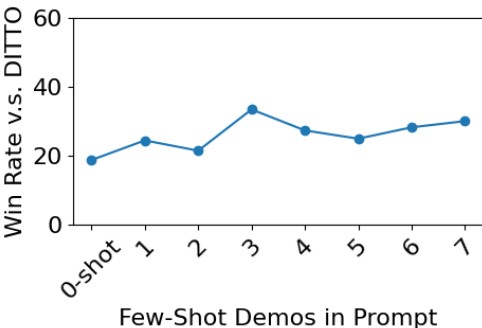

Figure 5: **Ablations for the number of demonstrations in few-shot prompted GPT-4.** We report win-rate vs. DITTO for a varying number of demonstrations in the few-shot prompt. While increasing the number of demonstrations in the prompt is positively correlated with improved performance, win rates are well under 50% and improvements are non-monotonic, with notable variance as we continue adding demonstrations.

In addition, when pairing GPT-4's zero-shot outputs to the target author's texts with the same setup as above, we get an accuracy of $98.8 \pm 0.84\%$. This indicates that the human text is correctly considered as more stylistically consistent than GPT-4's output in most cases and provides evidence that our GPT-4-based evaluation setup is not overly biased towards its own outputs.

## G   USER STUDY DETAILS AND EXAMPLE DEMONSTRATIONS

### G.1   USER STUDY INTERFACE

Our interface consists of two parts: a data collection phase where we solicit tasks (Fig. 6) and demonstrations (Fig. 7) from users; and a preference elicitation phase (Fig. 8) where we ask individuals to select between pairwise generations across baselines.

### G.2   USER STUDY TASK GENERATION PROMPTS

While users can provide their own tasks as part of the data collection phase, we also generate prompts to make the process easier. Below, we outline the prompt for task generation. We replace {domain} and {audience} based on the input from the user.

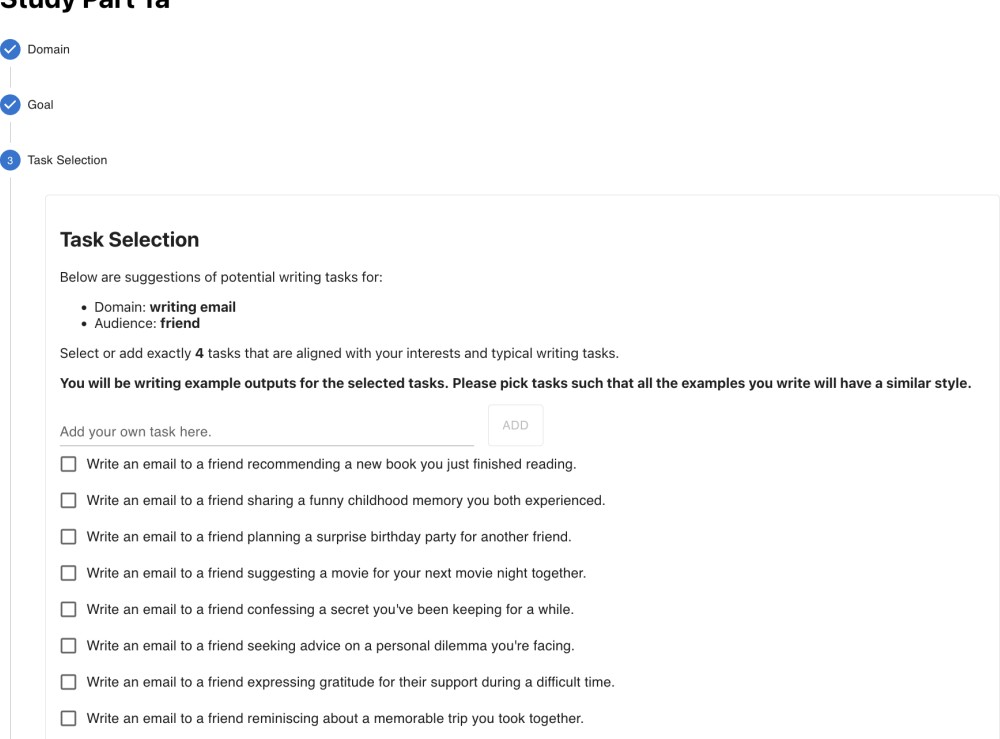

Figure 6: Task Elicitation Screenshot from the User Study. Individuals can either select GPT-4 generated prompts, or write their own.

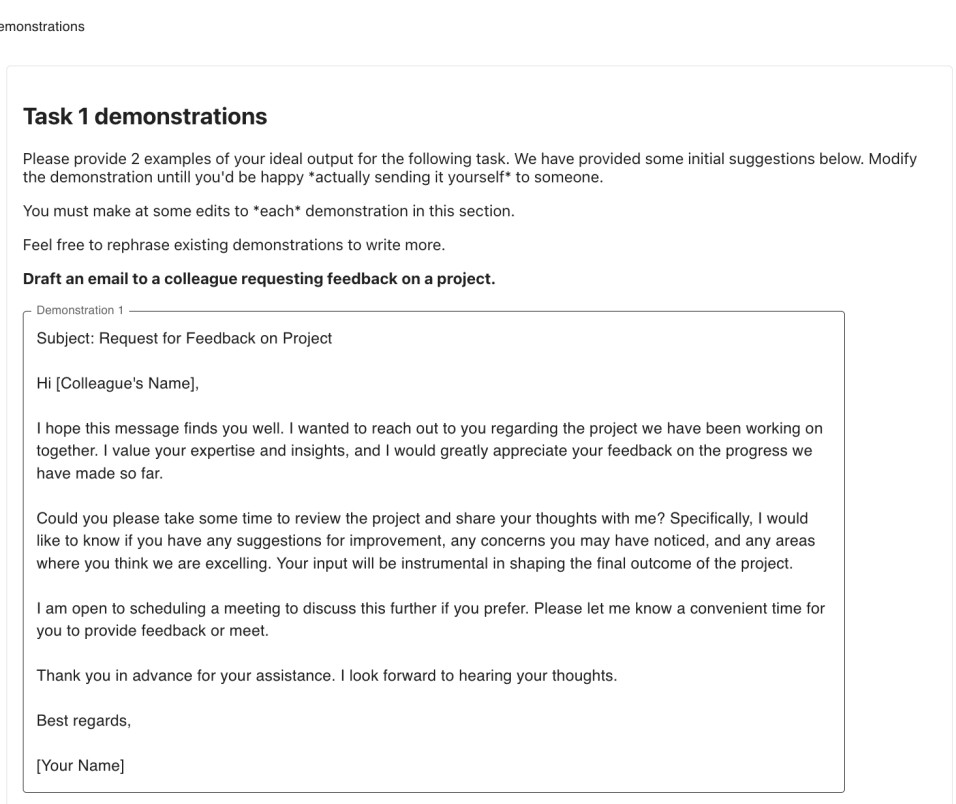

Figure 7: Demonstration Collection Screenshot. In this step, users provide demonstrations for 2 of the 4 selected prompts. We give GPT-4 zero-shot completions so that users can edit or rewrite to their liking. In the screenshot above, we show the GPT-4 completion before edits.

**Text Evaluation Tasks**

**Task 1: Compose an email proposing a sponsorship package tailored to the interests of a potential sponsor from Berkeley High.**

For the following 10 questions, please select your **preferred** text example based on the **task** listed above

**①** Q1

**Please select the text example that you prefer:**

**Example A**

Subject: Customized Sponsorship Opportunity for Berkeley High

Dear [Name],

I hope this email finds you well. I am extremely excited to propose a sponsorship opportunity that aligns perfectly with Berkeley High's mission and engages your school community's interests.

We are offering a Silver Sponsorship for your school's partnership. This sponsorship comes with the following benefits:

1. Branding Recognition: Gain prominence as a Silver Sponsor, including recognition on the event website and during our live event.

2. Complimentary Booth Space: Display your school's offerings, resources, and merchandise at our event for the community to explore and engage with. This is an excellent opportunity to connect with families and local businesses.

**Example B**

Subject: A Unique Blend of Power Partnerships - Collaboration Proposal from Berkeley High

Hey there!

Greetings from the Berkeley High Hawks family. I hope this email finds you, your team, and your loved ones in the best of health and spirits.

Given our shared passion for education, youth, and local community engagement, I am incredibly excited to propose an opportunity that we believe fits perfectly with our mutual interests.

As you may know, Berkeley High is home to a vibrant community of over 3,000 students who represent a diverse, creative, and active population of tomorrow's leaders. Within our school, we have several initiatives, clubs and sporting teams that regularly make a mark and bring commendation to our institution.

Figure 8: Pairwise Preference Screenshot. In the final step, users are asked to rank 20 pairwise preferences comparing all training paradigms.

```
Please generate {n_tasks} ChatGPT writing prompts for the domain of "{domain}" and an
audience of "{audience}".

EXAMPLES: Here are some examples of writing prompts based on a domain of "Writing
email" and an audience of "personal friends":

Write an email to a friend inviting them to dinner.
Write an email to a friend asking for advice.
Write an email to a friend sharing a recent accomplishment.

Like our examples, make sure the new tasks you generate will result in writing with the
same style or values (e.g.  all casual, all formal, all honest, etc.)

Respond with ONLY a JSON with the following format:
{
  "tasks":  [
    "<Task 1>",
    "<Task 2>",
    "<Task 3>",
    ...
  ]
}
```

### G.3 EXAMPLES OF DEMONSTRATED FEEDBACK

We include some examples of demonstrated feedback to highlight the types of changes users make.

#### G.3.1 USER A

***Train Prompt***

Write an email to a friend discussing a recent movie or TV show you watched.

***Original GPT-4 Generation***

Subject: Let's dish about the latest movie we watched!

Hey [Friend's Name],

I hope you're doing well! I wanted to chat with you about the movie I just watched last night - "Inception". Have you seen it yet? It's a mind-bending thriller that kept me on the edge of my seat the whole time.

The storyline is so unique and complex, with layers of dreams within dreams that kept me guessing until the very end. The visuals and special effects were absolutely stunning, and the acting was top-notch. Leonardo DiCaprio was fantastic in his role, as always.

I don't want to give too much away in case you haven't seen it yet, but I highly recommend it if you're into thought-provoking films that make you question reality. Let me know if you've seen it and what your thoughts are - I'd love to hear your take on it!

Looking forward to catching up with you soon.

Take care, [Your Name]

***User's Demonstration***

Subject: WTF even happens in Tenet?

I know you like Inception, but your boy Nolan's newest movie is Tenet and OMG I have no idea what happens. Like they are traveling through time, but not like normal time travel, they just move backwards through time. Like everything happens as if a movie were played in reverse. So, like, when people who are moving forward in time (the way we do in real life) see the people who are moving backward in time they actually look like they're moving in reverse. And then kenneth branagh is like somehow communicating with the future and building a weapon to kill everyone for unknown reasons. I have watched it three times and I literally still have no idea what happens. But the special effects are so dope I just keep rewatching it, lol. N-E-WAYZ watch it and tell me what you think mmmm-kay?

---

***Test Prompt***

Write an email to a friend sharing a new recipe you tried and loved.

***DITTO Output***

Hey friend! Hope you're doing well. This weekend I tried out this new vegan recipe for lentil bolognese (lentil bolognesa?) and it was amazing!! If you try it, the sauce is actually made with lentils instead of ground meat. You have to cook it down for like 45 minutes (you can even make it the night before and let it sit) — it's the most hands off thing, and it thickens the longer it simmers. Then just toss it with pasta, garlic bread and you're good to go. Lentils are the best for filling in the place of ground meat, and this felt so satisfying as a meal. So I highly recommend, it's on all the vegan cooking instagrams but there's a million ways to make it [...]

### G.3.2 USER B

**Task Prompt**

Email a faculty member requesting guidance on preparing an effective research mentorship application.

**Original GPT-4 Generation**

Subject: Request for Guidance on Research Mentorship Application Preparation

Dear [Faculty Member's Name],

I hope this email finds you well. I am reaching out to seek your guidance and advice on preparing an effective research mentorship application.

I am interested in applying for a research mentorship opportunity [specific program or opportunity if applicable] and I believe that your expertise and experience could greatly enhance my application. I have a strong interest in [briefly mention your research interests or specific area of study] and I am excited about the prospect of working with you to further develop my research skills.

I would greatly appreciate it if you could provide me with insights on what makes a successful research mentorship application, including key elements to focus on, potential strategies to highlight my strengths, and any other advice you may have based on your experience in this field.

I am committed to dedicating the necessary time and effort to submit a competitive application and value your guidance in this process. Your support and mentorship would be invaluable to me as I strive to further my research goals and academic pursuits.

Thank you in advance for considering my request. I look forward to your advice and suggestions to help me craft a strong research mentorship application.

Warm regards,

**User's Demonstration**

Subject: Request for Guidance on Research Mentorship Application Preparation

Dear [Faculty Member's Name],

My name is [my name], a PhD student at [university name]. I am applying to [name of this faculty member's mentorship program] for Summer 2024. I am excited about the possibility of participating in your institute's program, and I am reaching out with a question about faculty participating in your program.

My research interests and prior experience span several areas: [briefly mention your research interests in a concise list or sentence]. However, these areas are relevant to several faculty within your institute – [list 2-3 names] – some of whom may not be advising students this summer through your program.

If possible, could you please let me know if any of these faculty are participating in your program in summer? I would love to apply if any of these faculty are accepting students.

Thank you!

---

**Test Prompt**

Write an email requesting mentorship opportunities for an undergraduate research position.

**DITTO Output**

Dear [Name],

I am a student at [University's Name], and I am searching for research mentorship opportunities this summer. I recently came across [Research Club/Organization's Name] and was excited to see the variety of programs you offer for undergraduates. I was hoping you could provide some guidance on which programs are still accepting applications, and their respective deadlines. I am particularly interested in any program that offers research opportunities, as well as stipends.

Thank you in advance for your help! If someone could let me know, I'd be very grateful.

### G.4 Demonstrations for Sample Efficiency Task

**Task Prompt**

Write an email informing lab mates that we will be having ice cream this weekend as a lab social.

***Demonstration #1***

We are gonna get some EYE SCREAM this weekend at [place] for our social. It's getting really friggin hot. Plus, you know, me and ice cream...

Whenever you get time: can you reply to me ASAP so I can have a good idea of what the count looks like? I'll send some more details in a bit re time.

See ya'll there!

[Name]

***Demonstration #2***

ATTENTION!!! VERY URGENT!!

Ice cream this weekend at [place]. We haven't had a social in a bit; plus [person] is gonna join us too.

Lemme know if [time] works for you all! If not, we can figure something else out.

Be there or be a melted ice cream cone,

[Name]

---

**Task Prompt**

Write an email informing students that there will be no seminar next week.

***Demonstration #1***

Hey folks!

We won't be having a seminar this week. Let me know if you have any questions for next week, though!

[Name]

***Demonstration #2***

Hi everyone!

Just a reminder that there won't be a seminar this week. See you next week! As always, feel free to reach out if you have any questions about the seminar in general.

[Name]

## H Supplementary Experiments

### H.1 Mitigating DITTO Forgetting

We additionally evaluated DITTO on HumanEval Chen et al. (2021) with a randomly sampled author ($a_{10}$) on CMCC. DITTO-ed models are specialized to a specific individual, so we expect some degradation. We can proactively mitigate degradations by selectively dropping DITTO's LoRA adapter, routing instructions between the general instruction-following model (Mistral 7B) and the specialized LoRA adapter (ala MoE). To route queries, we experimented with the following zero-shot prompt, prompting the general model.

When routing queries, we observe no degradation—our pass@1 remains the same (0.31). In other words, our prompted router perfectly identifies which tasks are suitable for the adapter. Without this mitigation, we do observe significant degradation ($0.31 \rightarrow 0.13$).

```
I have a specialized model trained on data of the form:

{demonstrations}

Should I use the specialized model or a more general-purpose model for the following
task?

{human_eval_task}

Respond with just SPECIALIZED or GENERAL.

Answer:
```

| Word | Log Odds | Z Score | P Value |
|---|---|---|---|
| appreciate | 1.40 | 2.82 | < 0.000 |
| greatly | 1.07 | 2.44 | 0.015 |
| time | 0.72 | 2.44 | 0.015 |
| understanding | 1.35 | 2.42 | 0.016 |
| message | 2.45 | 2.35 | 0.019 |
| effectively | 1.67 | 2.17 | 0.030 |
| trust | 2.27 | 2.16 | 0.031 |
| initiatives | 1.27 | 1.94 | 0.052 |
| thought | 2.05 | 1.93 | 0.054 |
| words | 2.05 | 1.93 | 0.054 |

Table 6: **Words more associated with few-shot GPT compared DITTO** in our user study. GPT relies on cliches and flowery language ("hope this finds you well"), even after few-shot prompting. Users found cliches hard to eliminate completely with prompting alone.

## H.2 DEMONSTRATION COHESIVENESS AND DITTO PERFORMANCE

To evaluate if demonstration cohesiveness affects DITTO performance, we prompted an LLM to score the cohesiveness of a set of demonstrations. We used scoring prompts from Lam et al. (2024), a system for LLM-based document clustering. We then computed Pearson's R correlation coefficient between cohesiveness scores (1 - 5 likert scale) and performance increases compared to the few-shot baseline. We found a moderate positive correlation ($R = 0.42$) between the cohesiveness of author demonstrations and the downstream performance.

One could automatically cluster a large set of documents into specific sub-styles; and then train DITTO models individually on each cluster. Since LLM-judged cohesiveness correlates with downstream performance, automatically assembling a set of DITTO adapters from a training corpus is a potential avenue for future work.

| Dataset | | |
|---|---|---|
| CMCC | CCAT50 | User Study |
| *Write an approximately 500 word essay to the following prompt: The city of Denver has decided to legalize small amounts of marijuana for persons over 21. How do you feel about this?* | *Write an article that starts with the following: Two of Japan's most troubled banks took action on Tuesday to end their long-running financial problems, with one finding a merger partner and the other finalising a radical restructuring.* | *Write an email to a friend sharing a new recipe you tried and loved.* |
| **Author-written** | | |
| Marijuana is one of those things that's still stigmatized for few legitimate reasons, and most of the reasons people feel a certain way about it are also true of alcohol. One huge, bone-crushingly important, monumental difference between the two is this: the government can and does effectively tax the manufacture and distribution of alcohol (and tobacco, for that matter). Denver has taken a step in the right direction, and maybe the only step that needs to be taken. Since people are given permission to carry only small quantities, theoretically this still legislates against large-scale wheeling and dealing while preventing cops from having to divert time and attention to minor drug busts, which are often a waste of resources [...] | [...] The moves are seen as just the beginning of a drive by the the indebted banking industry to get ready for Japan's emerging "Big Bang" era of financial deregulation. Hokkaido Takushoku Bank, Japan's 10th-largest commercial bank, said it will merge with the smaller Hokkaido Bank Ltd and cease its overseas operations to become a "super-regional" bank. Such super-regionals use their local power base to build a large presence that rivals their big-city competitors. Meanwhile, Nippon Credit Bank Ltd, one of three long-term credit banks in Japan, was expected to hold a news conference later in the day on a restructuring that would include an injection of government money and bankruptcy filings [...] | Note: samples from the user study have different train prompts. We have no ground truth for the test prompts.

*Write an email to a friend planning a virtual movie night.*

Guess what we are doing on Saturday!?!? Having a virtual movie night!!!

Here are the details:
Date: [Date] [...]

I will send you the link later. Tell Graham to come too. I've invited Bob and Sarah so I think we'll have a good crew. Do not bail!! [...] |
| **DITTO** | | |
| The Denver City Council recently voted to approve the legalization of small amounts of marijuana for residents 21 and over (The Denver Post). I absolutely support this decision. This is a move towards common sense law. It saves the police significant amounts of time and resources. I'd argue that the enforcement of all laws concerning drugs should take this angle. It's also beneficial for other reasons (revenue to the city, taxpayer dollars spent more effectively, etc) Opponents frequently argue for the "children being present in the streets" theory. Fine, add a proviso [...] | [...] Under that plan, the government vowed to fully privatise Japan's troubled financial institutions by the 1998 business year. "The government will help with the mergers and will provide a stable environment in which the institutions can take steps toward becoming self-sustaining," Finance Minister Ryutaro Hashimoto told reporters after a meeting on Tuesday with executives of the two troubled institutions. The two had already taken the first necessary steps toward their "dying business" scenarios, in which they have almost no chance of standing on their own [...] | Hey friend! Hope you're doing well. This weekend I tried out this new vegan recipe for lentil bolognese (lentil bolognesa?) and it was amazing!! If you try it, the sauce is actually made with lentils instead of ground meat. You have to cook it down for like 45 minutes (you can even make it the night before and let it sit) — it's the most hands off thing, and it thickens the longer it simmers. Then just toss it with pasta, garlic bread and you're good to go. Lentils are the best for filling in the place of ground meat, and this felt so satisfying as a meal. So I highly recommend, it's on all the vegan cooking instagrams but there's a million ways to make it [...] |
| **Few-shot GPT-4** | | |
| I am certainly intrigued by the city of Denver's recent decision to decriminalize small quantities of marijuana for individuals who have reached the legal drinking age. Given the myriad of implications this policy shift has for government, society, and individuals alike, it is certainly a development worthy of thoughtful examination. [...] | [...] In what is seen as a significant development in the banking industry, stricken Sumitomo Trust & Banking firm announced a merger deal with established Sakura Bank Ltd. Meanwhile, Tokai Bank Ltd, grappling with similar issues, unveiled a comprehensive restructuring plan aiming to nurse the bank back to health over the next few years [...] | Subject: YOU NEED TO TRY THIS RECIPE!!! OMG! You have GOT to hear about this bomb-diggity recipe I just discovered! It's a pasta dish from Italy's homemade spaghetti aglio e olio!! YES, it sounds fancy but trust me it's super simple, idiot-proof, and the finished product, it's lip-smackingly DELISH! The main stuff you need are spaghetti, garlic, olive oil, chilli flakes (for the kick), and EVERYONE's fav cheese Parmesan [...] |
| **SFT** | | |
| The Mile High City has made headlines once again as the result of an ordinance passed earlier this year. Denver, Colorado's main metro area, has made history by legalizing small amounts of marijuana for persons over the age of 21. The Denver Police Department now cannot levy charges against those found in possession of the substance in sizes considered "reasonable"; the law, however, bans the sale and consumption of marijuana in any publicly accessible vicinity. [...] | [...] Yamatane Seimei First Real Estate Co Ltd will be the first major institution to be wound up in Japan since World War Two, when Japan's wartime debt was liquidated, the firm said in a statement. The firm said it will request on Wednesday that the government order it into a special public liquidation scheme – which is supposed to be a last resort for rescued financial institutions. [...] | Hey friend, So last night I had some people over for dinner, and I cooked this amazing chicken recipe. I'm not gonna lie, it took a little bit of work, but it was totally worth it. It was a skillet chicken marsala, and it tasted ah-mazing. I cooked the chicken in butter until it was golden brown on the outside, and then I took it out and cooked some shallots and mushrooms in the pan. Then I added some marsala wine, chicken stock, and cream, and let it simmer for about 20 minutes. At the very last, I added the chicken back to the pan to cook through while the sauce reduced. |

Table 7: **Selected prompts and responses across datasets (CMCC, CCAT50, User Study) and methods (DITTO, SFT, Few-shot GPT-4) for all evaluated models.** All generations are produced on unseen prompts drawn from the test set. Compared to the author-written ground truth, we observe that SFT occasionally feels longwinded, or fails to capture quirks of the author's writing. Meanwhile, few-shot GPT prompting yields outputs that sound "GPT-like" (*myriad of implications, significant development, etc.*), or over-does the author's original style (i.e. is cringe). Qualitatively, DITTO generations best preserve the author's voice and style.

