# OpenReview forum: "Aligning Language Models with Demonstrated Feedback"
_ICLR.cc/2025/Conference — ICLR 2025 Poster_

### Official Review · Reviewer_HDJC · 2024-10-20

**Soundness:** 3
**Presentation:** 4
**Contribution:** 3
**Rating:** 6
**Confidence:** 4

**Summary:**

This paper proposes an alternative to RLHF which is effective at learning from a few demonstrations. The paper shows that this method outperforms supervised finetuning and few-shot learning. The paper shows human eval results, qualitative samples, and various quantitative evals to show that DITTO is effective at getting models to adapt to a new task based on a few examples. The paper also discusses the connection between DITTO and imitation learning, explaining why the method might outperform just using supervised learning (as is common in LLM work) to do imitation learning, and why you might even expect to get better performance than the existing examples. The algorithm basically works by using the LLM to generate examples that are assumed to be worse than the demonstrations, then constructing pairwise preferences between the LLM generated samples and the expert demos (and possibly between earlier vs. later LLM checkpoints in the training run), then using DPO to learn from the constructed pairwise ranking.

**Strengths:**

-The method outperforms few-shot learning, which is surprising/impressive to me, I didn't expect that and it was one of my main doubts about the method from just reading the abstract. I think this could be a pretty compelling method potentially for doing automated red teaming, where you'd want to match some target data distribution as closely as possible, in order to elicit the most representative behavior from the model you're red teaming. This could then help with eliminating backdoors or sleeper agents (https://arxiv.org/abs/2401.05566), which is probably the application of this that I think most stands out to me as different from what is covered from prior work (I'm not that aware of many effective supervised learning alternatives like DITTO)
-The method seems useful for settings where fine-tuning an existing RLHF model (though I'm a bit less clear how broadly this would work / if this would replace RLHF for finetuning across lots of tasks or just some specific ones related to adapting the model's style or writing)
-Well-written paper, easy to follow
- The approach itself is clever, and it's interesting/surprising to me that it works well
-Nice that there are some human eval results, those helped to convince me that there are real gains with the method over few-shot learning (where it's clear the model hasn't adapted its behavior much).
-Likewise, the samples in the appendix are quite helpful for the above too
-Analysis in Table 3 is great for explaining why this might work
-Section 5 analysis is great/helpful.
Connecting DITTO to imitation learning is helpful for explaining why this is interesting, and why it would work.

I would give this paper a 7/10 rating, somewhere between marginal accept and accept (but the form would only allow a 6 or an 8).

**Weaknesses:**

-Would be most compelling if evaluated on higher expertise tasks: like coding complex tasks or forecasting. Seems like one of the main areas of relevance, given that this is where we might expect to be in the low-data regime where we want to get the most of our a small amount of (high-quality or hard to obtain) data. I also expect it to be harder/more impressive to see gains in these domains. Currently, the tasks are fairly basic and all writing related. Enough for a proof-of-concept but probably not complex enough to make me want to use DITTO instead of RLHF everywhere.
-One of the most interesting applications of the method would be to get generalization beyond what the demos are able to provide, it would be very compelling if this method led to generalization beyond the demos (which seems to be potentially possible if the method is working well, based on the discussion in the paper, if I understand correctly)
-The paper would ideally compare to Constitutional AI, another popular RLHF-alternative. (Though this could take some time to reimplement, if there aren't publicly available implementations). More generally, I'm unsure if the method outperforms using principles to guide/instruct the model (especially if those principles are derived by an LLM from the few examples, which would be most comparable to the existing method/setting). The results showing that prompting doesn't fix all the issues help here, but more sophisticated methods like Constitutional AI could still outperform DITTO here
- I'd love to see scaling trends on how well this works across model sizes -- it would be most compelling if the gains in task reward over supervised learning / few-shot learning seem to improve as models grow larger, rather than shrink
- I'm not sure but it's possible to me that this method partly beats few-shot learning on RLHF models because RLHF models are resistant to adaptation with few-shot examples, but that the method wouldn't outperform few-shot learning if using pretrained LLMs (or maybe even just instruction-tuned/supervised learning finetuned models). That could potentially be a helpful experiment to run (and more compelling if DITTO also outperforms other adaptation techniques when comparing on a pretrained language model)

Minor:
-Would be nice to show at least 1-2 examples in main paper, to show the sample quality. (Having these in the appendix is helpful though)
-The method could be explained more clearly sooner in the paper, I think that I didn't understand the actual algorithm until page 4 or so, when it would be nice to understand it from the intro or abstract itself

**Questions:**

Some questions I had while reading the paper (some might be out of scope for this paper or for the rebuttal period):

Does this work for 1-shot learning?
Do all fine-tuning runs use LoRA?
Does this work better for highly realistic/plausible synthetic data? Does this look indistinguishable to an LLM from some other real distribution, even after the LLM is fine-tuned? That would be a really compelling use case for this (to help with doing automated red teaming, with realistic looking inputs that closely match the target data distribution)
Does it help few-shot to explicitly instruct needed to be very close to few-shot examples in style? Or was that just tried for fitting zero-shot?
How do you choose hyperparameters with such a small number of examples? Like SFT/DPO ones? If you were doing any hyperparam selection, you might run into issues like described here: https://arxiv.org/abs/2105.11447
How did you pick the 20/80 data mix? How robust is that across datasets/settings?
How well does DITTO work in higher data regime? That would be the most compelling result, if it could replace RLHF when using large amounts of data (which is how it's often used in practice)

---

> ### Author Response · Authors · 2024-11-23
> **Response for HDJC (1/2)**
>
> Thank you for your comprehensive and careful review! We’re glad that you found DITTO’s performance impressive, and our method clever. We’re also glad you appreciate the user study and our lexical analysis. Finally, we appreciate the insightful application ideas (e.g. improved red-teaming) — we’ve included these suggestions in our future work!
>
> ### Generalization beyond the demonstration and tasks with more expertise
> We agree that this is an important application area! Please see the general response (see point 1). The TL;DR is that our current setup indeed covers generalization beyond niche tasks, but we’re not sure if DITTO can teach new complex reasoning skills—we’ve addressed this in our revised limitations section.
>
> ### Confounding effects of RLHF
> We agree that RLHF can have confounding effects, and we’ve introduced a new experiment to test this. See point 2 in the general response. The TL;DR is that general instruction following capabilities may be required as a “starting point” — jointly learning instruction-following and demonstrated feedback is too difficult a task to learn from a handful of demonstrations.
>
> ### Comparisons to Constitutional AI
>
> Even though Constitutional AI uses a small set of principles, it's still bottlenecked by the same issues of pairwise preferences. Consider the following algorithm.
>
> 1. Take demonstrations from the user and convert the demonstrations into principles.
> 2. Follow Constitutional AI:
>    1. Sample generations from the LLM.
>    2. Use the principles to:
>       - (a) Label pairwise preferences (with an LLM).
>       - (b) Train the model.
>
> If none of those generations at step 2.1 are close enough to the user's desired output, then the method will not succeed.
>
> We effectively tried an “upper bound” of this in Section 5.3, where a human annotated many pairwise preferences with an ideal set of demonstrations in mind. In other words, we used a human in step 2.1 instead of an LLM. This is similar in nature to Constitutional AI, where an LLM would instead annotate pairwise preferences. The big problem we ran into occurred when we sampled pairwise preferences from π_ref. We observed that generated pairs were out-of-distribution relative to the demonstrations—pairwise preferences do not reach a user’s demonstrated behavior. In other words, the samples generated by an LLM (2.1) are so far from the user’s ideal behavior that the samples from the LLM never get close, and the labeled LLM preferences (2.2b) are irrelevant. We think something similar will probably happen for Constitutional AI too. We added a few lines discussing this intuition in the updated version of section 5.3.
>
> ### Examples + Explain DITTO earlier.
>
> We're glad you liked the examples! We will definitely move some of the Appendix examples to the main text given the extra space — we’re working on a new figure for that. We’ve additionally revised the abstract, adding a few sentences to explain the high-level algorithm in more detail.
>
> > DITTO operates by having an LLM generate examples that are presumed to be inferior to expert demonstrations. The method iteratively constructs pairwise preference relationships between these LLM-generated samples and expert demonstrations, potentially including comparisons between different training checkpoints. These constructed preference pairs are then used to train the model using a preference optimization algorithm (e.g. DPO).

---

> ### Author Response · Authors · 2024-11-23
> **Response for HDJC (2/2)**
>
> We also really appreciated all your questions --- we've been mulling over similar ones for follow-up work. We couldn't carefully empirically evaluate all of them, but we are considering a FAQ in the final paper answering these questions. Regardless, we hope these responses help!
>
> **1-shot learning.** From our demonstration scaling experiments (Figure 2 and Sec 5.2), getting the model to learn demonstrated feedback might need at least three samples. Qualitatively, we’ve played around with DITTO-ed models on a single demonstration—there’s definitely a behavioral shift, but I think three is where you see generalization beyond the specific demonstrated behavior.
>
> **Finetuning with LoRA.** See general response (point 3)
>
> **Does this look indistinguishable to an LLM from some other real distribution?** Personally, we think the generations from DITTO look surprisingly “real.” Beyond our lexical analysis (Table 3), there are a bunch of odd linguistic quirks we see in the outputs (e.g. typos, coordinating conjunctions, sentences starting with “and”, etc.). We didn’t think of the red-teaming application, and we’ll definitely add it to the future work.
>
> **Does it help few-shot to explicitly instruct needed to be very close to few-shot examples in style? Or was that just tried for fitting zero-shot?** We tried explicitly instructing the few-shot variant too! Apologies if that was unclear—we’ve revised that line in our paper (models)
>
> **True few-shot learning.** Our setting is indeed a true few-shot setting. We have a withheld validation set that is approximately the same size as our actual test set! We agree that a much larger validation set, or engineering directly on the test set, would result in the effect documented by Perez et al. Additionally, our hyperparameter sweeps were done on a randomly selected author—still, DITTO generally outperforms all baselines across most authors where no tuning was done.
>
> **Model Scaling.** Unfortunately, there isn’t a Mistral model larger than 7B parameters; and moving all our experiments to a larger model was cost-prohibitive. We couldn’t test this hypothesis while keeping everything else fixed—we’ve mentioned this in the limitations.
>
> **Data mixes, splits, and higher data regimes.** We don’t know (yet). Unfortunately, there aren’t many datasets where many demonstrations come from a single user, and where we can ablate these properties. DITTO’s design choices are definitely focused on low-resource settings; applying our method to higher-resource settings will require additional algorithmic / engineering work!

---

### Official Review · Reviewer_41aT · 2024-10-27

**Soundness:** 2
**Presentation:** 3
**Contribution:** 3
**Rating:** 8
**Confidence:** 4

**Summary:**

The paper proposes a method called Demonstration Iterated Task Optimization (DITTO), designed to align large language models (LLMs) with user-specific preferences using a minimal number of user-provided demonstrations. This method eliminates the need for large-scale datasets typically required for supervised fine-tuning or RLHF. The paper claims that DITTO can significantly improve the alignment of LLMs for user-driven tasks and offers a practical solution for customizing language models. The paper explains their theoretical insights from online imitation learning with practical implementations, demonstrating effective customization for real-world applications like email writing and author-specific content generation.

I recommend accepting this paper, as it tackles a significant challenge and presents an interesting solution that is well-supported in theory and through empirical evidence. This method can have a strong impact on making LLMs more customizable and accessible. However, I strongly recommend that the author provide further empirical evidence that demonstrate the effectiveness of this method on more tasks/datasets - this would significantly improve the quality of this work.

Comments:
- The theoretical grounding in online learning is well-detailed and provides a clear explanation as to why the method works. The empirical validation further strengthens these theoretical claims.
- The proposed method is designed for practical applications. This is an important factor when applying LLMs in real-world situations.

Suggestions for improvement
- Consider expanding the evaluation to include a wider range of domains. Specifically, investigate tasks tasks that require general alignment rather than user-specific tasks. This would provide a clearer picture of DITTO’s versatility and scalability. I think even negative results would be very informative.
- It would be helpful to include a more detailed analysis of how the quality of demonstrations impacts performance. This could include testing DITTO with intentionally ambiguous or low-quality demonstrations to assess robustness.
- The limitations section could be expanded with a deeper discussion on the trade-offs of using few-shot demonstrations. Exploring scenarios where the approach might fail or require adjustments would strengthen the paper’s transparency.
- A more granular analysis of failure cases would add depth to the evaluation. This could involve detailed case studies highlighting scenarios where the method struggles.

**Strengths:**

- DITTO introduces a new approach to user-specific alignment by using a small set of demonstrations to generate online comparison data. This is innovative and practical for settings where data collection is costly.
- The paper provides a strong theoretical justification for DITTO, grounding it in online imitation learning. The derivation explains why DITTO can outperform traditional methods like SFT in low-data scenarios.
- The paper completes various experiments, demonstrating DITTO’s effectiveness across static benchmarks (e.g. email writing, news articles) and in a user study. The method consistently outperforms traditional techniques like few-shot prompting and SFT, providing convincing empirical support.
- The authors have made the code accessible, allowing for others to reproduce and validate their results

**Weaknesses:**

- Limited exploration is done into how DITTO scales to broader and more diverse tasks that may require a more generalized alignment. This is seen in how the experiments primarily focus on a small number of demonstrations.
- DITTO’s approach heavily relies on the quality of user-provided demonstrations. If demonstrations are unclear or poorly constructed, the alignment could suffer. This could limit DITTO’s real-world applicability when high-quality demonstrations are not readily available.
- The paper primarily focuses on text-based tasks. However, it would be interesting to understand the effectiveness of DITTO’s method in aligning LLMs in other modalities or more complex reasoning situations.

**Questions:**

- How does the method scale with larger LLMs, and are there specific challenges in aligning models that have stronger RLHF priors?
- How does DITTO perform in broader tasks that require more generalized alignment rather than user-specific customization? Could you provide insights into its scalability beyond niche tasks?
- How sensitive is DITTO to the quality of demonstrations? Could you elaborate on strategies to mitigate the impact of poorly constructed or ambiguous demonstrations?
- In terms of computational efficiency, how does DITTO compare with existing approaches when scaling to larger datasets or more complex tasks?

---

> ### Author Response · Authors · 2024-11-23
> **Response for 41aT**
>
> We thank 41aT for their thorough and thoughtful review! We're glad that 41aT thinks DITTO "tackles a significant challenge and presents an interesting solution that is well-supported in theory and through empirical evidence"; and that our work has "strong impact on making LLMs more customizable and accessible." Below, we address your questions:
>
> ### Wider Range of Domains and Modalities
> See the general response (point 1). The TL;DR is that while our current setup indeed highlights generalization beyond niche tasks, we would not expect that DITTO to teach new reasoning skills that haven’t been seen by the the pretrained LLM. We also do indeed observe some forgetting on general alignment post-DITTO-ing on specific demonstrations, but we propose a prompt-based routing mitigation that addresses this (see general response, 1.2).
>
> ### Sensitivity to Demonstrations
> This is a great point!
>
> We ran an experiment to see if performance improvement compared to the few-shot baseline was correlated with “demonstration cohesiveness.” We prompted an LLM to score the cohesiveness of demonstrations, modifying prompts from [1] on LLM-based document clusterer. Then, we computed Pearson’s R correlation coefficient between cohesiveness scores (1 - 5 likert scale) and performance increases. We find a moderate positive correlation (R = 0.42) between the cohesiveness of author demonstrations and the downstream performance.
>
> One could automatically cluster a large set of documents into specific sub-styles; and then train DITTO models individually on each cluster. Since LLM-judged cohesiveness correlates with downstream performance, automatically assembling a set of DITTO adapters from a training corpus is a potential avenue for future work. **We’ve referenced this analysis in our limitations and have outlined a new section in the Appendix.**
>
> [1] Lam et al. 2024. Concept Induction: Analyzing Unstructured Text with High-Level Concepts Using LLooM
>
> ### Confounding Effects of RLHF
> We agree that RLHF can have confounding effects, and we’ve introduced a new experiment to test this (see general response, point 2). Please see the general response. The TL;DR is that general instruction following capabilities may be required as a “starting point” — jointly learning instruction-following and demonstrated feedback is too difficult a task to learn from a handful of demonstrations.
>
> ### Computational Efficiency
> See the general response, point 4. TL;DR is that our primary bottleneck is in sampling, but work on faster inference (e.g. VLLM) can easily mitigate this. We’ll add this to the limitations and future work.

---

### Official Review · Reviewer_yVim · 2024-11-02

**Soundness:** 3
**Presentation:** 3
**Contribution:** 3
**Rating:** 5
**Confidence:** 4

**Summary:**

This paper identifies a key issue: current LLMs, aligned to represent the collective voice of many, often fail to align specifically with any individual preference due to contradictions among them. While guiding LLMs toward a general preference is feasible, it requires substantial preference data. The authors propose a method, DITTO, to align LLMs to specific settings using fewer than 10 demonstrations drawn from existing interaction logs or direct edits to LLM outputs. These demonstrations are treated as "golden" examples, while outputs from current and previous LLM checkpoints are rejected. Through author attribution tasks and user studies, they demonstrate the effectiveness and sample efficiency of DITTO.

**Strengths:**

The paper introduces DITTO, a novel method designed to guide LLMs toward specific settings for effective customization, achieving sample efficiency with fewer than 10 demonstrations. DITTO outperforms strong baselines, including SFT and GPT-4 with few-shot prompting. Additionally, a detailed user study further reinforces the reliability of DITTO.

**Weaknesses:**

1. The static experiments in Section 4.1 are not particularly convincing. Have you considered testing additional baselines or employing other automatic evaluation methods, such as calculating sentence embedding similarity to compare styles?
2. Have you evaluated DITTO on more benchmarks or tested its generalization ability? I noticed that only three authors were used for validation or testing. Can the DITTO method generalize to tasks beyond writing?

**Questions:**

1. In Section 3, you introduce the core method of DITTO and compare it with online imitation learning. What is the purpose of Section 3.3?
2. How did you determine the percentage distribution of the paired data, specifically the 70% online data, 20% replay data, and 10% intermodel pair data?
3. In Table 1, for the CMCC dataset, why do the zero-shot and few-shot results from GPT-4 appear the same in column a9, both at 40.28%? Additionally, why do both SFT and DITTO show results of 81.94% without any improvement? How would you comment on this?

---

> ### Author Response · Authors · 2024-11-23
> **Response for yVim**
>
> We thank yVim for their thoughtful and thorough review! We appreciate that yVim appreciates our user study, our focus on few-shot alignment, and our strong performance improvements over available baselines.
>
> ### Metrics and Static Benchmarks are Not Convincing
>
> _Metrics_
>
> Beyond just GPT-eval, we did try both sentence embeddings and perplexity measures. We abandoned both for performance reasons. We found that both perplexity and sentence embeddings did not discount degenerate outputs. Repetitions of phrases that appear in a generation result in inflated scores from both PPL and sentence embeddings [1]. Our observation—on degenerate text yielding low PPL—is already a well-documented finding (see [1]). While we reference these reasons in 4.1 (automatic evaluation) we will explicitly mention alternative metrics, namely embeddings and perplexity, and the associated challenges. We will also cite related work on text degeneration and its relationship to perplexity. We additionally validated GPT-eval in Appendix F.2, and found that it was quite good at judging authorship (98\% accuracy).
>
> [1] Holtzmann et al. 2020. The Curious Case of Neural Text Degeneration
>
> _Beyond static benchmarks_
>
> We agree that GPT-eval is not perfect, so we spent a significant amount of time working on a user study to complement our static benchmarks. Many of the tasks are quite diverse in nature—from writing recipes to asking for advice. In addition, we sourced preferences from each user: our setup ensured that the user writing the demonstration also evaluated their own DITTO-ed model. We document many of the provided demonstrations in the Appendix. In the final paper, we will move a handful to the main text.
>
> ### Generalization Ability
>
> To clarify: we have ten authors for the test set, not three. Our splits are not done at the author level, but at the demonstration level. Each author has 7 train demonstrations, ~3 validation demonstrations, and ~3 test demonstrations. For each author a_{1…10} we train a DITTO model on 7 demonstrations, and then validate / test on ~3. All of the results in Table 1 are done on the test split. We understand that Table 5 in the appendix is confusing, and we’ve revised the caption to be more explicit.
>
> In terms of generalization beyond writing, please see the general response (1.1 and 1.3).
>
> ### What’s the purpose of Section 3.3?
>
> In Section 3.2 we derive and present the DITTO method intuitively, demonstrating how by treating the demonstrations as "gold data" that we can use to generate more preferences. In comparison to prior work, we make several design choices (e.g., a constant reference policy) that lead to improved performance in the low-data regime, per our results in Table 1.
>
> Section 3.3 is designed to complement the intuitive explanation in 3.2 with a more formal grounding in an imitation learning / RL perspective. Specifically, we demonstrate that the intuitive choice of treating demonstrations as preferred to model samples has a mathematical grounding in the area of *online imitation learning*.  Specifically, DITTO's objective can be viewed as optimizing the min-max Max-Ent IRL game popularized by [2]. The result is a more theoretical verification of our design choices to complement our empirical results.
>
> If you have any further questions about this, we are happy to answer!
>
> [2] Ziebart et al. 2008. Maximum entropy inverse reinforcement learning.
>
> ### How did you determine the percentage distribution of the paired data, specifically the 70% online data, 20% replay data, and 10% intermodel pair data?
>
> This is a hyperparameter we optimized in the hyperparameter search setup. Apologies for the oversight—we included this in the **revised version of Appendix D (Hyperparameters).** To summarize, we tried a handful of setups with varying amounts of paired data. In general, we qualitatively observed that online and replay data comparisons were most stable, and intermodal comparisons less-so. Increasing intermodal percentages beyond 30\% resulted in degenerate output.
>
> ### What’s going on with Author 9?
>
> We looked into A9’s demonstration data to identify some reasons. First, A9’s demonstrations stylistically vary a lot from task to task; and second, A9’s opinions are bit… polarizing (re: A9 has fairly conservative opinions on gay marriage and religious freedoms). We suspect that few-shot GPT doesn’t improve because A9’s opinions are likely in conflict with the values encoded in GPT-4.
>
> As for the tied performance between SFT and DITO: A9’s demonstrations are qualitatively quite different from one another. We suspect that DITTO is especially useful when demonstrations are more cohesive. To mitigate this, one could cluster demonstrations beforehand and train DITTO on individual clusters. **We ran some preliminary experiments on demonstration sensitivity and our clustering approach (Appendix H), and have mentioned sensitivity in the revised limitations.**

---

> > ### Comment · Reviewer_yVim · 2024-11-23
> >
> > ## Metrics and Static Benchmarks
> >
> > We appreciate the authors' comprehensive analysis using both sentence embeddings and perplexity measures. The observation that repeated phrases in the generated text can inflate scores from both PPL and sentence embeddings is insightful. However, we would welcome further discussion on why GPT-4-based evaluation appears to be more robust against these degenerated phrases. Additionally, it may be worth exploring whether the occurrence of repeated phrases could be attributed to specific generation hyperparameters, such as repetition_penalty and temperature settings.
> >
> > ## Generalization Ability
> > We value the authors' efforts in general responses. To strengthen the claims about DITTO's generalization ability, we would suggest expanding the evaluation scope in several areas. For the author attribution task, considering a broader range of authors during both training and evaluation could provide more compelling evidence. While the current examples in general responses effectively illustrate certain capabilities, a more comprehensive evaluation across diverse scenarios would be beneficial. Regarding generalization to domains like code and mathematics, incorporating established reasoning benchmarks could help validate the model's practical applicability in real-world human conversations.

---

> ### Author Response · Authors · 2024-11-24
> **Thanks for the reply!**
>
> Thank you for the reply! Taking a crack at your observations:
>
> > Further discussion on why GPT-4-based evaluation appears to be more robust against these degenerated phrases.
>
> There've been a few papers that study LLM-based evaluation. These papers compare against other metrics / human evaluation, and find that LLM-based evaluators often align with human evaluation more than other metrics. One reason for doing so is handling degenerate outputs---LLM-based metrics evaluate outputs more as a whole, according to prior work. We've cited a handful of papers that support GPT eval in general below. We'll include this discussion in the revised paper (under automatic evaluation, Section 4.1).
>
> Zheng et al. 2023. Judging LLM-as-a-Judge with MT-Bench and Chatbot Arena. NeurIPS
>
> Chiang et al. 2023. Can Large Language Models Be an Alternative to Human Evaluations? ACL
>
> Dubois et al. 2023. AlpacaFarm: A Simulation Framework for Methods that Learn from Human Feedback. NeurIPS
>
> Kim et al. 2023 Prometheus: Inducing fine-grained evaluation capability in language models. ICLR
>
> > Exploring whether the occurrence of repeated phrases could be attributed to specific generation hyperparameters, such as repetition_penalty and temperature settings.
>
> We did explore repetition_penalty ablations, but at this point our models were already too overfit to a specific demonstration, memorizing and repeating sentences from the train demos. Model outputs would not generalize to new tasks. In our setting, we think GPT-eval captured both of these failure modes and was likely the best option. Still, the gold standard is human-eval; the user study in our paper additionally aligns with our results.
>
> > While the current examples in general responses effectively illustrate certain capabilities, a more comprehensive evaluation across diverse scenarios would be beneficial. Regarding generalization to domains like code and mathematics, incorporating established reasoning benchmarks could help validate the model's practical applicability in real-world human conversations.
>
> We agree! However, established reasoning benchmarks do not have gold standard or high-quality demonstration-based data---mostly just questions and answers. While we would have loved to include more writing domains and authors, unfortunately there aren't many sources of gold-demonstrated feedback---just model-generated (potentially unfaithful!) CoTs. In fact, for our paper, we had to repurpose author attribution datasets for our evaluation. Creating a new benchmark for demonstrated feedback would be a wonderful avenue for future work! For example: collecting a reasoning dataset across which gold v.s. subpar trajectories are carefully labeled, or a writing dataset with orders of magnitude more demonstrations per author. We're happy to outline this in the revised paper.
>
> Thank you again for engaging! We hope our rebuttal addressed most of your concerns :) Let us know if you need anything else!

---

### Official Review · Reviewer_7eqT · 2024-11-04

**Soundness:** 3
**Presentation:** 3
**Contribution:** 2
**Rating:** 6
**Confidence:** 4

**Summary:**

The paper introduces a novel method, Demonstration Iterated Task Optimization (DITTO), for training large language models (LLMs) with expert demonstration datasets in a more data-efficient manner. Through a mathematical derivation, the authors illustrate how DITTO functions as a form of online imitation learning. They validate the method's effectiveness by utilizing a GPT-4 evaluation scheme and compare it against several other approaches, including Supervised Fine-Tuning (SFT), SPIN, and few-shot prompting. The authors conclude that DITTO is particularly advantageous for training LLMs to adopt specific writing styles or user preference tuning, outperforming other methods in these areas.

**Strengths:**

- The paper proposes a data-efficient training method that enables LLMs to follow expert demonstrations. The Reinforcement Learning from Human Feedback (RLHF) data can be continuously generated by simply comparing expert demonstrations with the intermodel's responses. This approach can also be seen as a blend of Reinforcement Learning from AI Feedback (RLAIF) and RLHF, making it a reasonable and effective method.
- The authors demonstrate the performance improvements of DITTO-trained models using GPT-4 evaluation and validate the method's effectiveness through a large-scale user study.
- They provide a theoretical perspective on the connection between online imitation learning and demonstrate that online imitation learning can outperform Supervised Fine-Tuning (SFT). The mathematical derivation and explanations are clear, and the results are further supported by meticulously designed ablation studies.

**Weaknesses:**

- The authors did not investigate potential side effects, such as performance degradation on other benchmark datasets, after training with DITTO. Since the LLM is fine-tuned exclusively on targeted demonstrations, there’s a risk of significant performance drops in broader tasks. It is essential to preserve the LLM's original knowledge and abilities while adjusting its output to align with specific style and preference.
- Also they overlooks the computational inefficiency of iterative training in an online imitation learning framework. This process requires substantial time and GPU resources, as it involves initializing the policy 𝜋0 (equivalent to SFT), generating responses from 𝜋0, training with DPO, and then iterating to produce 𝜋1, and so forth. These steps are difficult to reproduce and demand more computational power than SFT baseline. Furthermore, achieving faster response generation in the trained LLM would require additional engineering efforts. Although DITTO improves data efficiency, it is also crucial to consider computational efficiency, given the high costs of training and generating responses with LLMs.
- The authors did not explore the limitations of the DPO algorithm or other potential approaches for training LLMs in a Reinforcement Learning from Human Feedback (RLHF) framework. It is known that the DPO algorithm can pose risks when training on preference datasets, as it may forget data from the "winning" side due to inherent mathematical issues.

**Questions:**

- Do you think DITTO would be effective for the coding skills or mathematical problem solving skills of an LLM?
- Have you attempted training the LLM without LoRA, using full fine-tuning instead?
- What kind of source code is used to generate online responses? If you were to train a much larger LLM (such as LLAMA 72B), would it be feasible to apply the online imitation learning method in the same way?

---

> ### Author Response · Authors · 2024-11-23
> **Response for 7eqT**
>
> We thank 7eqT for their thoughtful review! We appreciate that 7eqT thought our work was particularly advantageous for writing and user-specific finetuning (we agree!); and that our approach is an effective blend of RLHF and RLAIF. Below, we address your questions:
>
> ### Handling Performance Degradations in General Alignment
>
> See the general response (point 1.3) — we introduce a new experiment to evaluate this! TL;DR: while we do observe degradations on tasks unrelated to what we train DITTO on, we can proactively mitigate this by selectively routing/dropping the LoRA adapter.
>
> ### Computational Efficiency
>
> See the general response (point 4). TL;DR is that our primary bottleneck is in sampling, but work on faster inference (e.g. VLLM) can easily mitigate this. **We’ve added this to the limitations and future work.**
>
> ### Limitations of DPO
>
> DITTO’s overarching setup is agnostic to the specific “*PO” optimization method. One could swap out DPO with KTO, ORPO, SimPO, etc. etc. We did some early experimentation with alternatives. In our setting, we observed no statistically significant difference—in practice—across the specific PO method. We’ve revised section 3.2 to note this! There may be some way to make training more efficient with reference-free approaches (e.g. SimPO), but we wanted to make sure our approach worked with vanilla DPO first. We leave this exploration to future work.
>
> ### Do you think DITTO would be useful for coding or math?
>
> Potentially! Please see the general response (point 1.1).
>
> ### Have you tried full finetuning?
>
> Yes! Please see the general response (point 3). TL;DR is that we observe no significant difference between LoRA, and we stuck with LoRA to save money / compute.
>
> ### How are you generating online responses?
>
> Right now, we’re using vanilla Huggingface code (see the TRL repository) to generate online responses—an anonymous repo of how we do this is in the paper (https://anonymous.4open.science/r/demonstrated-feedback-3531/). There are some tricks TRL employs with LoRA significantly that reduce memory usage: because we’re only finetuning the adapter as $\pi_{t}$, we do not have to save a separate reference model in memory—we can just disable the adapter and run a forward pass.
>
> In general, we think our codebase could be adapted to train much larger models. Our codebase’s bottleneck is primarily at inference. While we use FlashAttention, applying recent work on speeding up inference further improve performance (see vLLM). **We’ve revised the future work/limitations sections to address this.**

---

### Author Response · Authors · 2024-11-23
**General Response (1 / N)**

# General Response

We thank all the reviewers for taking the time to review our work, and for the thoughtful and thorough feedback! In particular, we appreciate that the reviewers valued DITTO’s novel approach to collecting and generating preferences through demonstration, our method’s theoretical grounding, and our focus on the practical customization of LLMs with limited feedback. Additionally, reviewers appreciated our meticulous ablations, our user study, and our qualitative analysis of examples.

While we address each reviewer’s feedback, we want to address shared questions around 1. generalization, 2. RLHF priors, 3. finetuning methods (LoRA v.s. full) and 4. efficiency.  All changes are in the revised manuscript (and are marked in blue).

Please let us know if you have any other questions- we're happy to answer followups!

## 1. Generalization Ability - All Reviewers

All reviewers raised questions regarding the generalization abilities of DITTO, asking if  DITTO-ed models extrapolate beyond the training demonstrations. We focus on three main points.

1.1 Will DITTO’s approach generalize to coding or reasoning tasks?

1.2 What about forgetting general capabilities / alignment?

1.3 Models must non-trivially generalize to perform well on our static benchmarks.

### 1.1 Will DITTO’s approach generalize to coding or reasoning tasks?

We would not expect DITTO to improve a model’s reasoning abilities. DITTO helps “focus” a model’s output distribution towards a specific demonstrated behavior, relying on online imitation learning for iterative improvement. In our writing evaluations, we suspect the model was trained on samples of writing similar to the author—DITTO iteratively brings this distribution to the fore. If the LLM was not pre-trained on many samples of expert reasoning, we doubt this distribution could surface; however, we don’t expect overall performance to worsen either (see 1.3).

Unfortunately, we are also unaware of pre-existing datasets with demonstrated feedback over complex reasoning problems (GPQA [1], for example, focuses on MCQ and not open-ended responses). We’ve discussed these concerns more explicitly in our revised limitations / future work.

We do, however, think DITTO would help with style for coding or reasoning problems. Aligned LLMs often generate verbose explanations or highly commented code. Programmers often have a stylistic preference when it comes to writing code. Here’s a qualitative example where we trained DITTO on a handful of coding demonstrations where we minified code.

Prompt
```
Write python code to read a JSON file.
```

Mistral Instruct
```
To read a JSON file in Python, you can use the json module. Here's a simple example:

import json

# Open the JSON file
with open('your_file.json', 'r') as f:
    # Load JSON data from file
    data = json.load(f)

# Now you can work with the data
print(data)

In this example, replace 'your_file.json' with the path to your JSON file. The json.load() function reads the JSON data from the file and returns it as a Python object (e.g., dictionary or list).
```

DITTO:
```
JSON.parse(require('fs').readFileSync(f.json'));
```

Even for coding tasks, we expect DITTO to have some impact on model behavior. **We’ve mentioned these reflections in our updated future work section.**

[1] Rein et al. 2023. GPQA: A Graduate-Level Google-Proof Q&A Benchmark

---

> ### Author Response · Authors · 2024-11-23
> **General Response (2 / N)**
>
> ### 1.2 What about forgetting general capabilities/alignment?
>
> Prompts from both our user study and our static benchmark (from the same author) are quite diverse, varying from writing recipes to asking for advice. Within this range, we observe no degradation. Still, all of these domains are related to writing.
>
> We suspect that the reviewer is interested in domains like coding, so we additionally evaluated DITTO on HumanEval [2], using a randomly sampled author (a_10) from CMCC. We can easily mitigate degradations by selectively dropping DITTO’s LoRA adapter, and routing instructions between the general instruction-following model (Mistral 7B) and the specialized LoRA adapter (ala MoE). We experimented with the following zero-shot prompt, prompting the general model.
>
> ```
> I have a specialized model trained on data of the form:
>
> {demonstrations}
>
> Should I use the specialized model or a more general-purpose model for the following task?
>
> {human_eval_task}
>
> Respond with just SPECIALIZED or GENERAL.
>
> Answer:
> ```
>
> This approach completely mitigates degradation.
>
> If one tries to use a specialized writing model for mathematical reasoning tasks, we would expect degradation: performance on HumanEval drops significantly for a DITTO-ed model (Instruct 0.31 -> DITTO 0.13).
>
> | Model                                | Pass @ 1 |
> |-------------------------------------|----------|
> | Mistral 7B Instruct                 | 0.31     |
> | DITTO                               | 0.13     |
> | DITTO + Prompted Router             | 0.31     |
>
> Finally, we updated the limitations section in the revision to be more explicit about the effects of forgetting, and we’ve added a section in the Appendix that outlines our mitigation approach. We think routing requests to specialized, demonstration-aligned models is a very interesting avenue for future work! While our prompted approach works, there are likely faster, more accurate, and more general methods.
>
> [2] Chen et. al 2021. Evaluating Large Language Models Trained on Code.
>
> ### 1.3 Models must non-trivially generalize to perform well on our benchmark.
>
> Within-author demonstrations span a diverse range of tasks, from both our static benchmarks and user study. DITTO-ed models must perform non-trivial generalization to perform well on our provided tasks. The submitted paper did not highlight this sufficiently. Here, we want to highlight the diversity of tasks in our author attribution benchmarks. Here are a handful of train-test prompts that highlight differences—we’ve included them in the Appendix.
>
> ```
> train: Discuss a recent movie or TV show you watched
> test: Share a new recipe you tried and loved.
>
> train: The city of Denver has decided to legalize small amounts of marijuana for persons over 21. How do you feel about this?
> test: Do you feel the Catholic Church needs to change its ways to adapt to life in the 21st Century?
>
> train: Write an email to your professor seeking advice on research topics for an upcoming project.
> test: Outline an agenda for a project meeting with a new collaborator.
>
> train: Share personal writing rituals and habits for inspiration.
> test: Highlight a fellow writer's work and encourage support within the community.
> ```
>
> To summarize, these tasks span opinion pieces, blog posts, recipe writing, requests to meet, etc. Performing well on these benchmarks requires non-trivial generalization. Across these, DITTO-ed models generalize substantially across different train/test prompts and topics, extrapolating from a very limited number of demonstrations and domains. **We’ve revised portions of our dataset (Section 4.1 and Appendix C) and user study (section 4.2) to make this remark more explicit!**
>
> ## 2. RLHF Priors (HDJC and 41aT)
>
> One observation shared by reviewers HDJC and 41aT is that our evaluated models are already instruction-finetuned and have strong RLHF priors—our baselines might be stronger on just base LLMs.
>
> To test this, we evaluated few-shot prompting and SFT on the base mistral model and compared to DITTO on CMCC. We found that even when using the few-shot prompted/finetuned base model, DITTO still significantly outperforms baselines.
>
> | Model                     | Win Rate v.s. DITTO |
> |---------------------------|---------------------|
> | DITTO                     | 50.0               |
> | SFT on Base Model         | 9.4                |
> | Few-shot on Base Model    | 10.4               |
>
> We suspect that general instruction-following capabilities are required as a “starting point” — jointly learning instruction-following and demonstrated feedback is too difficult a task to learn from a handful of demonstrations.  **We’ve included this analysis in Section 5.1.**

---

> ### Author Response · Authors · 2024-11-23
> **General Response (3 / N)**
>
> ## 3. Have you tried full finetuning or do you use just LoRA? (7eqT HDJC)
>
> We did try full finetuning for a few authors and noticed little / no change in performance (t-test). Since we didn’t observe much of a difference (and because full finetuning was significantly more resource-intensive across 20 models we needed to train) we stuck with LoRA for the entire paper. **We include a reference to this in Section 4.1 (models and baselines) of the revised paper.**
>
> ## 4. Have you considered computational efficiency? (7eqT and 41aT)
>
> Yes! In general, DITTO does take longer than SFT. DITTO is slower than training-free approaches (prompting) and SFT (15 minutes with DITTO vs. 2 minutes with SFT on 7 demonstrations). The largest bottleneck lies in sampling—our current implementation relies on vanilla HF code. However, we suspect a mix of prior (e.g., vLLM [25]) and future work in LLM inference optimization can improve DITTO’s speed. Once a DITTO model is fully trained, however, it yields a single LoRA adapter $\pi_n$ that has no inference overhead compared to any other adapter—we do not have to save any of the intermediate policies ($pi_0 … pi_{n-1}$).
>
> In addition, we are quite excited about inference time extensions of DITTO! We think that applying DITTO in-context—sampling negatives and using demonstrations as in-context feedback—is a promising approach. We’re leaving this as an avenue for future work; and have added an excerpt related to efficiency in the revised paper’s limitations section.

---

### Meta-Review · Area_Chair_YQp5 · 2024-12-22

**Metareview:**

## Summary
This paper introduces Demonstration ITerated Task Optimization (DITTO), a method for aligning language models to specific tasks using fewer than 16 demonstrations. Unlike methods like RLHF or supervised fine-tuning, which often require large datasets, DITTO leverages ideas from online imitation learning to align models efficiently. The approach constructs pairwise preferences between user-provided demonstrations and outputs generated by the model or its earlier checkpoints. These preferences are then used to guide training through a method like DPO. Experiments across tasks such as writing news articles, emails, and blog posts demonstrate that DITTO significantly outperforms alternatives like few-shot prompting and supervised fine-tuning, with a reported average improvement in win rates of 19 percentage points. A user study further supports the method’s effectiveness in customizing language model behavior.

## Decision

Overall, the paper provides a compelling contribution to aligning language models efficiently and effectively. The combination of theoretical grounding, empirical evidence, and practical utility justifies its acceptance.

The approach is grounded in online imitation learning, with clear theoretical derivations that explain why DITTO can outperform existing methods like supervised fine-tuning (SFT) in low-data settings. The method's connection to reinforcement and imitation learning is well-explained.

Extensive experiments demonstrate that DITTO outperforms strong baselines, including few-shot prompting and SFT, across multiple tasks such as email writing, news articles, and blog posts. The method shows an average improvement of 19 percentage points in win rates, validated through GPT-4 evaluations and a large-scale user study.

**Additional Comments On Reviewer Discussion:**

Overall, the reviewers were positive about this paper. Some of the reviewers, like Reviewer 7eqT, HDJC, and yVim, have raised important concerns about the scalability of the method, metrics, and evaluations. The authors have done a decent job addressing them overall. They have provided some additional evaluations. I recommend the authors incorporate the experimental results in response to reviewers' concerns into the final revision of the paper.

---

### Decision · Program_Chairs · 2025-01-22

Accept (Poster)